# Non-thalamic origin of zebrafish sensory nuclei implies convergent evolution of visual pathways in amniotes and teleosts

**Solal Bloch[1†‡], Hanako Hagio[2,3†], Manon Thomas[1§], Aurélie Heuzé[1], Jean-Michel Hermel[1], Elodie Lasserre[1], Ingrid Colin[1], Kimiko Saka[4], Pierre Affaticati[5], Arnim Jenett[5], Koichi Kawakami[4,6], Naoyuki Yamamoto[2], Kei Yamamoto[1]\***

[1]Paris-Saclay Institute of Neuroscience (Neuro-PSI), Université Paris-Saclay, CNRS, Gif-sur-Yvette, France; [2]Laboratory of Fish Biology, Graduate School of Bioagricultural Sciences, Nagoya University, Nagoya, Japan; [3]Institute for Advanced Research, Nagoya University, Nagoya, Japan; [4]Laboratory of Molecular and Developmental Biology, National Institute of Genetics, Mishima, Japan; [5]TEFOR Paris-Saclay, CNRS UMS2010, INRA UMS1451, Université Paris-Saclay, Gif-sur-Yvette, France; [6]Department of Genetics, SOKENDAI (The Graduate University for Advanced Studies), Mishima, Japan

**\*For correspondence:**
kei.yamamoto@cnrs.fr

[†]These authors contributed equally to this work

**Present address:** [‡]Department of Neuroscience, Charleston Alcohol Research Center, Medical University of South Carolina, Charleston, United States; [§]Plateau de phénotypage TEFOR, LPGP-INRA, Rennes, France

**Competing interests:** The authors declare that no competing interests exist.

**Abstract** Ascending visual projections similar to the mammalian thalamocortical pathway are found in a wide range of vertebrate species, but their homology is debated. To get better insights into their evolutionary origin, we examined the developmental origin of a thalamic-like sensory structure of teleosts, the preglomerular complex (PG), focusing on the visual projection neurons. Similarly to the tectofugal thalamic nuclei in amniotes, the lateral nucleus of PG receives tectal information and projects to the pallium. However, our cell lineage study in zebrafish reveals that the majority of PG cells are derived from the midbrain, unlike the amniote thalamus. We also demonstrate that the PG projection neurons develop gradually until late juvenile stages. Our data suggest that teleost PG, as a whole, is not homologous to the amniote thalamus. Thus, the thalamocortical-like projections evolved from a non-forebrain cell population, which indicates a surprising degree of variation in the vertebrate sensory systems.

## Introduction

It is accepted that all vertebrate brains possess three major divisions: the forebrain (prosencephalon), midbrain (mesencephalon), and hindbrain (rhombencephalon). The brain morphogenesis established along the anterior-posterior and dorsal-ventral axes of the neural tube occurs at early embryonic states (*Figdor and Stern, 1993*; *Joyner et al., 2000*; *Echevarría et al., 2003*; *Stern et al., 2006*; *Vieira et al., 2010*). Neuronal connectivity that determines the brain functions is established at later developmental stages. It is a fundamental question to evaluate to what extent the functional connectivity is conserved among different vertebrate groups (*Güntürkün, 2005*; *Yamamoto and Bloch, 2017*; *Striedter and Northcutt, 2020*).

This issue has been difficult to address, because connectivity patterns are often similar across vertebrate groups. Yet, there is no consensus on the regional homology of the related brain structures across species. This is illustrated by the still ongoing discussions about the evolutionary history of the sensory ascending pathways and thalamorecipient pallial areas in amniotes (a group containing mammals and birds). Notably in birds, several different thalamic nuclei convey sensory information to the pallium (Pal; dorsal telencephalon containing the cortex in mammals) in a modality-specific

manner. Thus, the avian thalamo-pallial pathway is often compared to the mammalian thalamocortical pathway (*Figure 1*; amniotes; Th → Pal) (*Butler, 1994a*; *Butler, 1994b*; *Reiner et al., 2005*). In contrast, homology of the target pallial areas has been under debate (*Karten and Shimizu, 1989*; *Bruce and Neary, 1995*; *Striedter, 1997*; *Puelles et al., 2000*; *Butler et al., 2011*; *Dugas-Ford et al., 2012*).

The principal problem is that it is difficult to determine the ancestral condition for the thalamo-pallial pathways of amniotes. Amphibians are the closest outgroup, but unlike amniotes, their sensory projections from the thalamus (Th) mainly terminate in the subpallium (SPa; ventral telencephalon containing the striatum; *Figure 1*; amphibians; Th → SPa), with little projections to the pallium (*Kicliter, 1979*; *Neary and Northcutt, 1983*; *Wilczynski and Northcutt, 1983*; *Butler, 1994a*).

Outside of tetrapods, brain structures are highly divergent compared to amniotes, and thus it is even more complicated to draw an evolutionary scenario. Although teleosts also have sensory afferents to the pallium, the morphology of the teleost pallium is very different from that of tetrapods, as the developmental processes are different (evagination in tetrapods versus eversion in teleosts). Different authors have proposed different hypotheses concerning which part of the teleost pallium would correspond to the mammalian neocortex (*Braford, 1995*; *Wullimann and Mueller, 2004*;

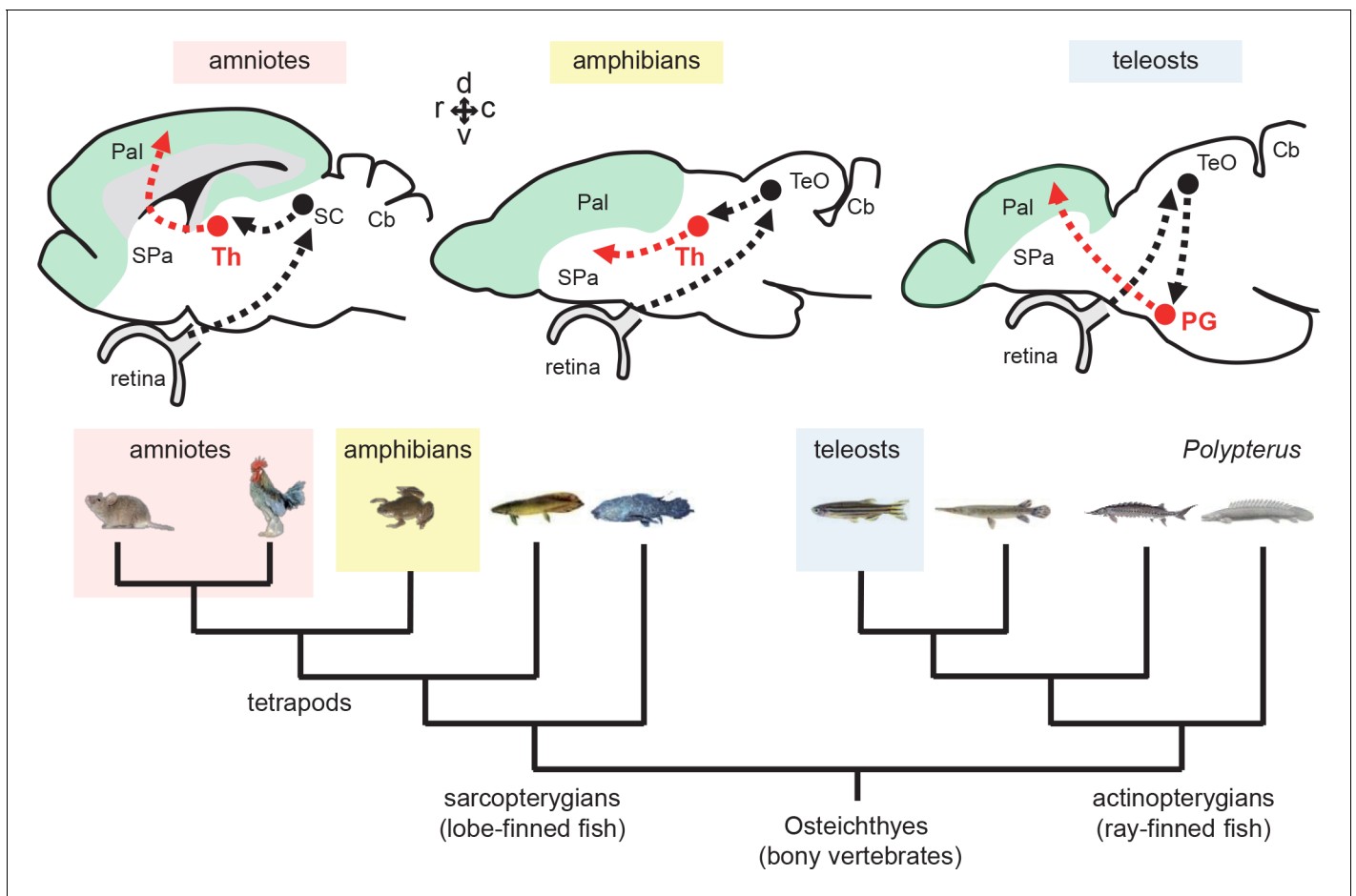

**Figure 1.** Tectofugal pathways in amniotes, amphibians, and teleosts and their phylogenetic relationships. Schematic drawings of sagittal sections of the rodent, frog, and zebrafish brains are shown above a phylogenetic tree of bony vertebrates (Osteichthyes). Note that tetrapods (including mammals, birds, and amphibians) and teleosts belong to two different groups of *Osteichthyes*: tetrapods are sarcopterygians (lobe-finned fish) whereas teleosts are actinopterygians (ray-finned fish). The ascending visual projections to the telencephalon are shown with red arrows. Although the connectivity patterns are similar, there are differences between taxa. The relay nuclei giving rise to the telencephalic projections are in the thalamus (Th) in tetrapods, while they are in the preglomerular complex (PG) in teleosts. The major telencephalic target is the pallium (Pal; indicated in green) in amniotes and in teleosts, while it is the subpallium (SPa) in amphibians. Abbreviations: Cb, cerebellum; Pal, pallium; PG, preglomerular complex; SC, superior colliculus; SPa, subpallium; TeO, optic tectum; Th, thalamus. Brain orientation: r, rostral; c, caudal; d, dorsal; v, ventral.

*Northcutt, 2006*; *Yamamoto et al., 2007*; *Yamamoto, 2009*; *Mueller et al., 2011*). Recent publications examining developmental processes of the zebrafish pallium have modified the classical view of the eversion model (*Folgueira et al., 2012*; *Dirian et al., 2014*; *Furlan et al., 2017*), and we have proposed a new interpretation on the pallial homology (*Yamamoto and Bloch, 2017*; *Yamamoto et al., 2017*).

In teleosts, the controversy is not only over the pallial homology. It has also been debated whether or not the structure giving rise to the major pallial projections is homologous to the tetrapod thalamus (*Wullimann and Rink, 2002*; *Northcutt, 2006*; *Yamamoto and Ito, 2008*; *Mueller, 2012*). This structure is named the preglomerular complex (PG; *Figure 1*; teleosts; PG → Pal). In order to determine the evolutionary scenario of sensory pathways in bony vertebrates (Osteichthyes), it is important to investigate whether or not the teleost PG is homologous to the amniote thalamus.

As for the controversy over the pallial homology, different authors claim different hypotheses. On the one hand, based on descriptive embryology, the teleost PG has been considered to originate from the basal portion of the diencephalon, the posterior tuberculum (*Bergquist, 1932*; *Braford and Northcutt, 1983*; *Butler and Hodos, 2005*; *Northcutt, 2008*; *Vernier and Wullimann, 2009*). In contrast, the thalamus originates from the alar portion of the diencephalon.

On the other hand, due to the similar connectivity patterns of sensory pathways, it has also been proposed that PG may be homologous to the thalamus (*Yamamoto and Ito, 2008*; *Ito and Yamamoto, 2009*). The PG receives various sensory inputs of different modalities (e.g. visual, auditory, lateral line, gustatory, and possibly somatosensory), and gives rise to ascending projections to the pallium (*Striedter, 1991*; *Striedter, 1992*; *Yoshimoto et al., 1998*; *Yamamoto and Ito, 2005*; *Yamamoto and Ito, 2008*; *Northcutt, 2006*; *Ito and Yamamoto, 2009*). Notably, the tectofugal visual pathway has been well investigated, and it has been compared to the one in amniotes. In goldfish and carp, the lateral preglomerular nucleus (PGl), a component nucleus within the PG, receives visual information from the optic tectum (TeO) and projects to the lateral part of the dorsal telencephalic area (Dl) within the pallium (TeO → PG → Pal; *Figure 1*; *Yamamoto and Ito, 2005*; *Yamamoto and Ito, 2008*; *Northcutt, 2006*). This is similar to the tectofugal or extrageniculate visual pathway in mammals: a visual thalamic nucleus (pulvinar in primates, and lateral posterior nucleus in other mammals like rodents) receives inputs from the TeO (superior colliculus -SC- in mammals) and projects to the visual pallium (extrastriate visual cortex in mammals) (SC → Th → Pal; *Figure 1*). In addition, the relative size and complexity of the PG are correlated with the complexity of the pallial connections, similarly to what has been documented in the tetrapod thalamus (*Northcutt, 2008*).

Some other studies have suggested that the PG consists of Pax6-positive (Pax6+) alar diencephalic cells that migrate ventrally during development (*Wullimann and Puelles, 1999*; *Wullimann and Rink, 2001*; *Wullimann and Rink, 2002*; *Mueller and Wullimann, 2002*; *Ishikawa et al., 2007*).

More recently, our cell lineage study in zebrafish has shown that many PG cells are migrating from the midbrain region, instead of the diencephalon (which is in the forebrain). The ventral brain regions containing the PG, the torus lateralis (TLa), and the inferior lobe (IL), which were considered to be part of the forebrain, are in fact composed of cells migrating from the midbrain-hindbrain boundary (MHB) (*Bloch et al., 2019*). These data strongly suggest that the PG is not, as a whole, homologous to the amniote thalamus. *Bloch et al., 2019* focused on the development of IL but left open the question whether the midbrain-derived cell cluster contains the PG-pallial projection neurons.

This study aims to clarify the developmental origin of the PG neurons comprising an ascending tectofugal visual pathway. Combining tract-tracing and cell lineage studies, we confirmed that the majority of the PG cells, including pallial projection neurons in the PGl, originate from the mesencephalic region. Thus, the teleost PG does not appear homologous to the amniote thalamus, implying that their similar connectivity patterns evolved independently in the two animal groups.

## Results

### Zebrafish transgenic line labeling the visual afferent projection to the pallium

In order to generate a transgenic line that labels the 'thalamocortical-like pathway' in zebrafish, we performed a genetic screen using the *Tol2* transposon-based gene trap construct, and collected transgenic fish with expression of the engineered Gal4 transcription factor in specific cell populations. The Gal4 expression was visualized by crossing the transgenic fish with a reporter line carrying EGFP under the control of UAS (*Tg(UAS:GFP)*) (*Asakawa and Kawakami, 2009*; *Kawakami et al., 2010*; *Lal et al., 2018*).

Among these lines, we identified a Gal4-expressing transgenic fish line, *Tg(gSAGFF279A)* crossed with *Tg(UAS: GFP)*, which had GFP-positive (GFP+) cells projecting to a part of the pallium (*Figure 2—figure supplement 1*). We analyzed the genomic DNA from the transgenic fish by Southern blot and inverse PCR, and found that the gene trap construct was integrated within an intron of the *inpp5ka* (inositol polyphosphate-5-phosphatase Ka) gene (*Figure 2—figure supplement 2*). Hereafter, we simply refer to this double transgenic line *Tg(gSAGFF279A;UAS:GFP)* as *Tg(279A-GFP)*, and always used the offspring screened with GFP expression.

In the *Tg(279A-GFP)*, abundant GFP+ fibers are present in the Dl (*Figure 2A*), and GFP+ cell bodies are found in a part of PG (*Figure 2B*). 3D reconstruction of confocal images of the entire brain allowed us to follow the projection from the PG to the pallium (*Figure 2C,D* and *Video 1*). We confirmed that axonal projections originating from the PG terminate in the ipsilateral Dl (*Figure 2C* and *Video 1*).

This projection from the PG to the Dl is quite similar to the ascending visual projection of other cyprinid species such as goldfish and carp (*Yamamoto and Ito, 2008*). In goldfish and carp, the vast majority of retinal axons terminate in TeO, and the TeO neurons project to the PGl, which in turn projects to the Dl in the pallium. In *Tg(279A-GFP)* zebrafish brain, GFP+ cells form a cluster within a nucleus corresponding to the goldfish PGl. Sagittal sections through the PG demonstrate the GFP+ cell cluster as a prominent oval-shaped structure (*Figure 2—figure supplement 1A*). Thus, based on the comparison with other cyprinids, we here refer to the GFP+ PG cell cluster projecting to the Dl as PGl.

The PGl-Dl projection is not yet observable at 4 weeks post-fertilization (wpf) (*Figure 3A,B*, and *Figure 3—figure supplement 1*). A few GFP+ cell bodies appear at the level of PG around 6 wpf. The GFP+ cell cluster in PG (which we consider as PGl) becomes prominent at 8 wpf (*Figure 3D*), yet the GFP+ fiber labeling in the Dl appear very weak at this stage (*Figure 3C*). At 3 months post-fertilization (mpf), there are around 200 GFP+ cells in the PG in each side of the brain (*Figure 2B–D*, *Figure 3F*, and *Figure 2—figure supplement 1I*), and they project heavily to the Dl (*Figure 2A,C,D*, *Figure 3E*, and *Figure 2—figure supplement 1A,B,F*).

### Validation of the tectofugal visual pathway in zebrafish

In order to verify whether the GFP+ projections can be regarded as a tectofugal visual pathway in zebrafish, we performed tract-tracing studies using DiI, biocytin, or biotinylated dextran amine (BDA).

It has been known that retinal projections terminate in the upper layers of TeO in a wide range of species of ray-finned fish (*von Bartheld and Meyer, 1987*), and it is also the case in zebrafish (*Figure 4—figure supplement 1*). In order to confirm whether the zebrafish PG receives visual inputs from the TeO, we injected DiI into the PG of the *Tg(279A-GFP)* line, targeting the lateral subdivision (PGl) with a guide of GFP (*Figure 4A*, asterisk). After 2–3 weeks of incubation, we observed retrogradely labeled cell bodies in a deep layer of TeO (*Figure 4B*). Each cell extends its dendrites up to the retino-recipient upper layer of TeO (the stratum fibrosum et griseum superficiale; SFGS), and this morphology is identical to the neurons receiving retinal inputs in carp and goldfish (*Yamamoto and Ito, 2008*). BDA injections into the TeO (*Figure 4C*, asterisk) labeled axon terminals in the PGl (*Figure 4D*, arrowheads), confirming that the PGl receives tectal inputs.

Following the biocytin injection into PGl (*Figure 4G*, asterisk), we observed abundant fiber labeling in the Dl (*Figure 4H*). This Dl labeling pattern is identical to the GFP+ fiber labeling of the *Tg (279A-GFP)* (*Figure 2A* and *Figure 4I*). Conversely, biocytin injections into Dl (*Figure 4J*, asterisk)

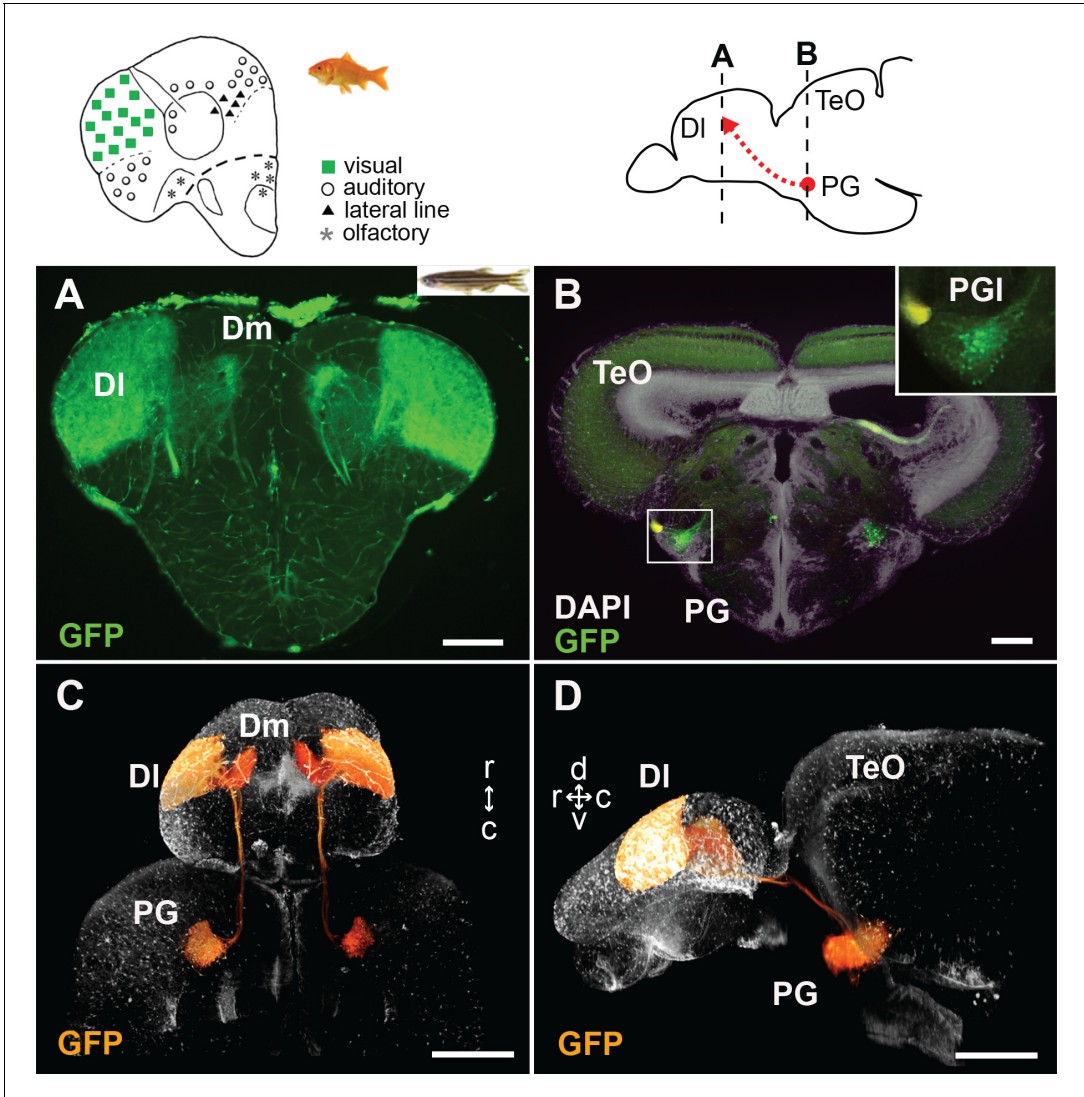

**Figure 2.** GFP+ afferents in *Tg(279A-GFP)* zebrafish transgenic line. (**A and B**) Frontal sections of *Tg(279A-GFP)* adult brain showing the GFP+ fibers in Dl of the pallium (**A**), and GFP+ cell bodies in PG (more specifically, PGl; **B**). The levels of the frontal sections are indicated on the schematic drawing of a lateral view of the brain (upper right). (**A**) Abundant GFP+ fibers are found in the zebrafish Dl, which corresponds to the main visual area in the goldfish pallium (upper left schematic drawing adapted from *Yamamoto, 2009*). (**B**) A brain section at the level of PG. Higher magnification of the right side of PG is shown in the inset. Based on the nomenclature from goldfish, we define the GFP+ PG cell cluster projecting to the Dl as PGl in the present study. (**C and D**) Selected visualization of the GFP+ projections from PGl to the pallium. After 3D reconstruction of the whole brain imaging of *Tg(279A-GFP)*, the GFP+ signal of the PGl cells was selectively visualized in order to follow their projections (see Materials and methods). **C** shows the dorsal view of the brain and **D** shows the lateral view. The original movie is shown in *Video 1*. Abbreviations: Dl, lateral part of dorsal telencephalic area; Dm, medial part of dorsal telencephalic area; PG, preglomerular complex; PGl, lateral preglomerular nucleus; TeO, optic tectum. Brain orientation: r, rostral; c, caudal; d, dorsal; v, ventral. Scale bars = 100 µm (**A and B**); 500 µm (**C and D**).

The online version of this article includes the following figure supplement(s) for figure 2:

**Figure supplement 1.** GFP expression of the *Tg(279A-GFP)* adult brain.
**Figure supplement 2.** Construction of *Tg(gSAGFF279A)* transgenic zebrafish.

labeled perikarya of PGl neurons (*Figure 4K*), coinciding with the position of the GFP+ cells in *Tg(279A-GFP)* (*Figure 4L*, arrowheads). These data confirm that the PGl neurons project to the Dl of the pallium.

Thus, we conclude that the PGl conveys visual information from the TeO to the Dl, and that the GFP+ projection from the PG to the Dl in *Tg(279A-GFP)* recapitulates this visual pathway.

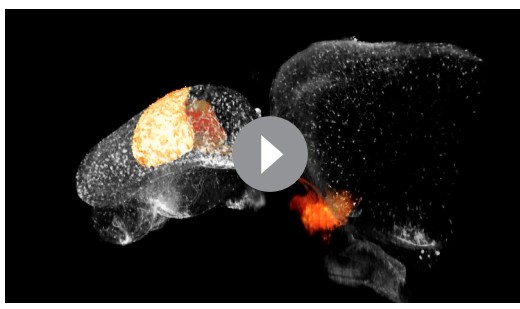

**Video 1.** 3D reconstruction of the PG-pallial visual afferents in the brain of the *Tg(279A-GFP)* transgenic line. Confocal images of the entire zebrafish brain were reconstructed to demonstrate axonal projections in 3D. Selective visualization of GFP+ signal by manual segmentation (shown in orange) demonstrates axonal projections originating from the PG neurons to the ipsilateral Dl.

https://elifesciences.org/articles/54945#video1

## Mesencephalic progenitors give rise to the GFP+ PG cells

Based on a cell lineage method using tamoxifen inducible Cre-lox recombination, we recently demonstrated that many cells of mesencephalic (midbrain) origin are present in some brain structures that have been considered to be of prosencephalic (forebrain) origin (*Bloch et al., 2019*). PG was one of them, thus we further investigated the development of PG cells, including those in PGl.

We used a double transgenic line generated by crossing *Tg(her5:ERT2CreERT2)* and *Tg (βactin:lox-stop-lox-hmgb1-mCherry)*, in which mCherry is expressed in the progenies of *her5*-expressing cells by a tamoxifen induction. The transcription factor *her5* is exclusively expressed in the midbrain-hindbrain boundary (MHB) at 24 hr post-fertilization (hpf) (*Figure 5A*), and in ventricular cell clusters in the midbrain (alar part such as the tectum) at juvenile stages (*Galant et al., 2016*; *Bloch et al., 2019*).

By following the mCherry-positive (mCherry+) cells at different developmental stages after induction at 24 hpf, we observe that PG is constituted as the most rostro-ventral end of a cell cluster migrating from the MHB (*Figure 5B,D,F* and *Video 2*). The *her5*-mCherry+ cells are distributed in the entire PG, including the PGl where the 279A-GFP+ pallial projection neurons are located (*Figure 5* and *Figure 5—figure supplement 1*). Thus, we conclude that many of PG cells, if not all, are of midbrain origin.

To further confirm whether the GFP+ and mCherry+ signals co-localize, we generated a quadruple transgenic line *Tg(her5:ERT2CreERT2;βactin:lox-stop-lox-hmgb1-mCherry;279A-GFP)* (*Figure 6—figure supplement 1*), and performed tamoxifen induction at 24 hpf. We have verified that the quadruple transgenic line is identical to the double transgenic lines in terms of expressions of GFP and mCherry, and that their brain development is unaltered. Observing the PG in the adult stage (3 mpf), we found that there are GFP+ PG cells co-expressing mCherry (*Figure 6C–F*, arrowheads). This suggests that at least some of the GFP+ pallial projection neurons originate from the MHB.

Since the GFP+ projection neurons become observable after 4 wpf (*Figure 3*), we hypothesized that they may be generated also during the juvenile stages. Thus, we decided to perform the tamoxifen induction at different developmental stages. However, the fertility of the *her5* quadruple transgenic line (number of eggs and survival rate of young larva) was relatively low. For this reason, we used an additional transgenic line for the induction at later developmental stages.

As an alternative to the *Tg(her5:ERT2CreERT2)*, we used *Tg(Dr830:ERT2CreERT2)* (*Heuzé, 2017*). The enhancer sequence '830' (human enhancer is named '*Hs830*', and zebrafish enhancer is '*Dr830*') is a highly conserved regulatory sequence in mouse and in zebrafish, which acts as a putative enhancer of the transcription factor *Meis2* (*meis2a* in zebrafish) selectively in the tectum (*Heuzé, 2017*). In this line, the expression territory of Cre is larger than *Tg(her5:ERT2CreERT2)* at 24 hpf, but the expression is limited to the tectal area after 30–48 hpf (*Figure 7* and *Figure 7—figure supplement 1*). We first generated the double transgenic line *Tg(Dr830:ERT2CreERT2;βactin:lox-stop-lox-hmgb1-mCherry)*, then crossed with *Tg(279A-GFP)*, in order to generate a quadruple transgenic fish *Tg(Dr830:ERT2CreERT2;βactin:lox-stop-lox-hmgb1-mCherry;279A-GFP)* (*Figure 8—figure supplement 1*).

We performed tamoxifen induction at different developmental stages from 24 hpf up to 8 wpf (*Supplementary file 1*), and examined the adult brains to verify whether GFP+ pallial projection neurons co-express mCherry. Similarly to the cases using *Tg(her5:ERT2CreERT2)*, tamoxifen treatments induced mCherry expression in many cells in the PG including the PGl. Counting mCherry+ cells

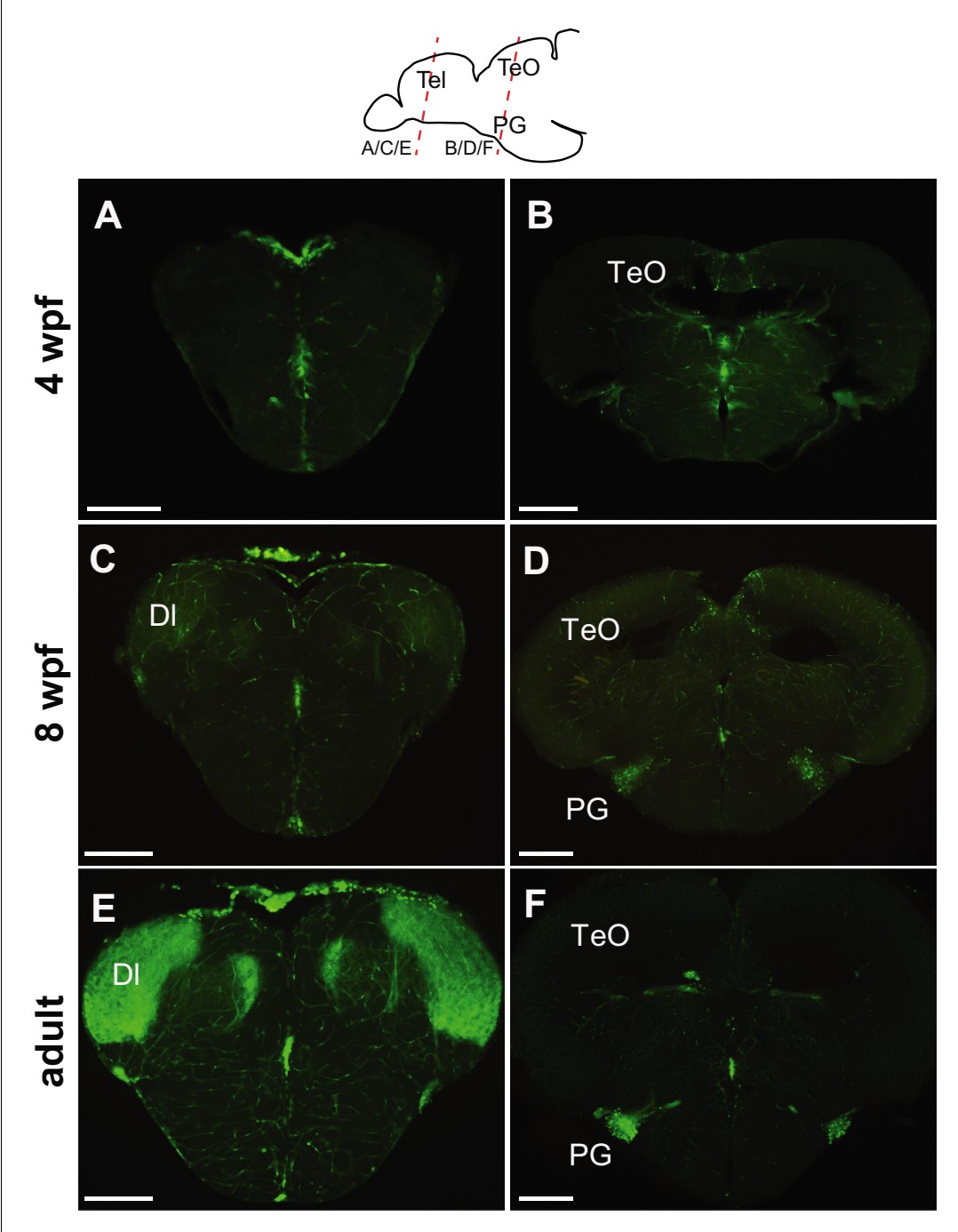

**Figure 3.** Progression of GFP expression in the *Tg(279A-GFP)* brain during development. Frontal sections of 4 wpf (**A and B**), 8 wpf (**C and D**), and the adult (**E and F**) brains at the level of the telencephalon (**A, C, and E**) and PG (**B, D, and F**). Approximate antero-posterior levels are indicated in the schematic drawing on the top. At 4 wpf, there is no GFP+ fiber in the Dl (**A**) nor GFP+ cell around the PG (**B**). The GFP+ cells (presumably in the PGl) become obvious at 8 wpf (**D**), but their fiber labeling in the Dl is significantly weaker (**C**) in comparison to the adult (**E**). Abbreviations: Dl, lateral part of dorsal telencephalic area; PG, preglomerular complex; TeO, optic tectum. Scale bar = 200 μm.

The online version of this article includes the following figure supplement(s) for figure 3:

**Figure supplement 1.** GFP expression of the *Tg(279A-GFP)* brain at 4 wpf.

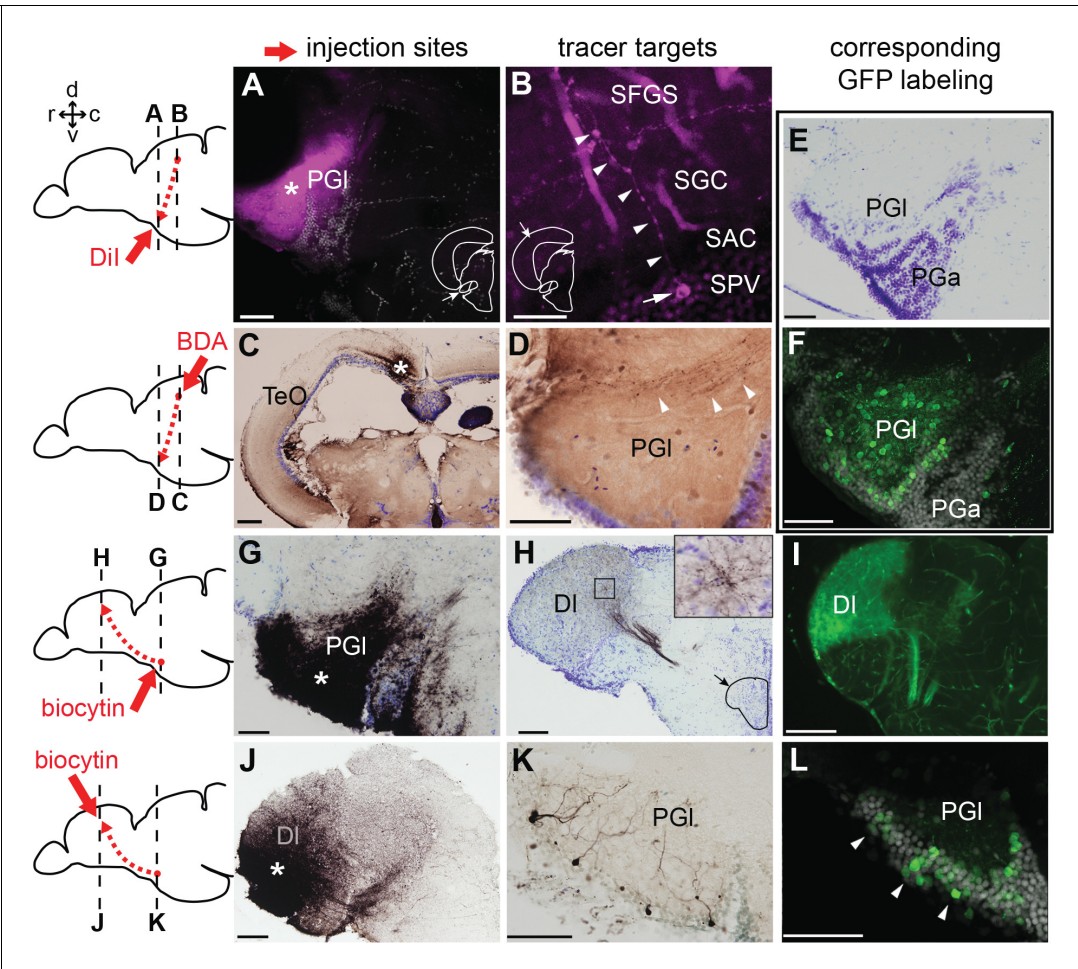

**Figure 4.** Tract-tracing study showing connections of the lateral preglomerular nucleus (PGl). Schematic drawings indicate the injection site of each tracer and levels of the frontal sections shown on the right panels. In all the frontal sections shown in A-L, the lateral side of the brain is to the left. (A–F) PGl receiving tectal inputs. (A and B) DiI retrograde labeling (magenta), showing the injection site in the PGl (A; asterisk), and a retrogradely labeled neuron (B; arrow) located in a deep layer (SPV) of the optic tectum (TeO). This neuron extends its dendrites (B; arrowheads) to the upper layer (SFGS) where retinal projections terminate. (C and D) BDA anterograde labeling (brown) showing the injection site in TeO (C; asterisk) and anterogradely labeled terminals in the ipsilateral PGl (D; arrowheads). (E) Cresyl violet staining showing the cytoarchitecture of the zebrafish PG: the lateral (PGl) and the anterior (PGa) subdivisions can be identified, based on the comparison with the goldfish PG (the nomenclature applied from goldfish; *Yamamoto and Ito, 2008*). (F) GFP+ labeling in the PG of *Tg(279A-GFP)* zebrafish line (20 µm projection of confocal images; GFP in green and DAPI in grey), showing a section comparable to the level shown in (E). GFP+ perikarya are mostly found in PGl. (G–L) PGl neurons projecting to Dl of the pallium. (G and H) Biocytin injection site in the PGl (G; asterisk) and anterogradely labeled terminals in the ipsilateral Dl (H). The right top inset in (H) shows a higher magnification of the squared area showing numerous labeled terminals. (I) GFP+ fiber labeling in the Dl of *Tg(279A-GFP)* zebrafish line, demonstrating an arborization pattern comparable to the anterograde biocytin labeling shown in (H). (J and K) Biocytin injection site in the Dl (J; asterisk), and retrogradely labeled neurons in the ipsilateral PGl (K) that extend dendrites ramifying in the neuropil. (L) GFP+ perikarya labeling in the PGl of *Tg(279A-GFP)* zebrafish line (arrowheads; 5 µm projection), demonstrating the almost identical cell localization as shown in (K). Abbreviations: Dl, lateral part of dorsal telencephalic area; PGa, anterior preglomerular nucleus; PGl, lateral preglomerular nucleus; SAC, stratum album centrale; SFGS, stratum fibrosum et griseum superficiale; SGC, stratum griseum centrale; SPV, stratum periventriculare; TeO, optic tectum. Brain orientation: r, rostral; c, caudal; d, dorsal; v, ventral. Scale bars = 50 µm (A, D-G, K, and L); 20 µm (B); 200 µm (C); 100 µm (H–J).

The online version of this article includes the following figure supplement(s) for figure 4:

**Figure supplement 1.** Retinal projections terminating in TeO.

from individuals induced at different time points suggests that at least 60% of PG cells would originate from the midbrain region (*Figure 8—figure supplement 2*).

We found GFP/mCherry co-expressing cells consistently at all the induction time-points until 6 wpf (*Figure 8*). The mCherry+ cells became less and less abundant along development. At 6 wpf, there were very few mCherry+ cells in PG, and we found only one cell co-expressing GFP and

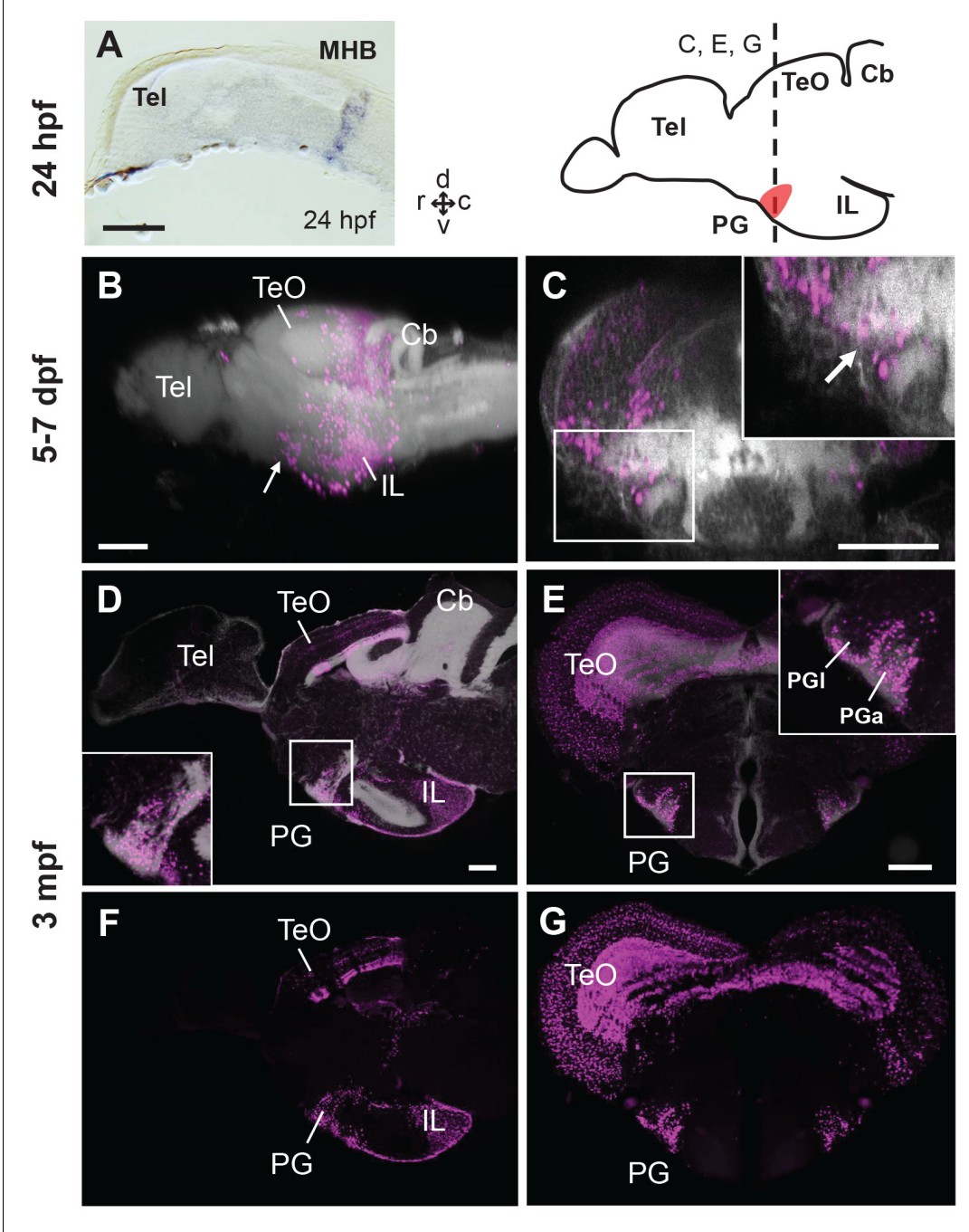

**Figure 5.** Progression of the mCherry+ cell distribution in the PG of *Tg(her5:ERT2CreERT2;βactin:lox-stop-lox-hmgb1-mCherry)* zebrafish treated with tamoxifen at 24hpf. (**A**) In situ hybridization (ISH) of *ert2Cre* showing that the expression of Cre in this line is limited to the midbrain-hindbrain boundary (MHB) at 24 hpf. (**B–G**) Sagittal (**B, D, F**) and frontal (**C, E, G**) views of brains showing the development of PG region. The schematic drawing of the zebrafish brain shows the position of PG (indicating the level of the frontal sections shown in C, E, and G). Higher magnifications of the PG region are shown in the insets. mCherry+ cells are shown in magenta, and brain morphology (DiD fiber labeling in B and C, DAPI in D and E) is shown in grey. (**B and C**) 3D reconstruction from confocal images of 5–7 dpf larval brains (images reused from *Bloch et al., 2019*). The arrows indicate the PG primordium. (**D–G**) Adult brain sections, with (**D and E**) and without DAPI (**F and G**). Abbreviations: Cb, cerebellum; IL, inferior lobe; MHB, midbrain-hindbrain boundary; PG, preglomerular complex; PGa, anterior preglomerular nucleus; PGl, lateral preglomerular nucleus; Tel, telencephalon; TeO, optic tectum. Brain orientation: r, rostral; c, caudal; d, dorsal; v, ventral. Scale bars = 100 μm.

*Figure 5 continued on next page*

*Figure 5 continued*

The online version of this article includes the following figure supplement(s) for figure 5:

**Figure supplement 1.** Distribution of mCherry+ cells in the PG following tamoxifen induction at 24 hpf in *Tg(her5: ERT2CreERT2;βactin:lox-stop-lox-hmgb1-mCherry)*.

mCherry among all the specimens examined (*Figure 8G*). We didn't observe any co-expression in the case of induction at 8 wpf.

Thus, our results suggest that GFP+ PGl cells are gradually added throughout the larval/juvenile stages around up to 6 wpf. Considering the small number of GFP+ cells in PGl and the short-term tamoxifen induction time, it would be reasonable to conclude that the majority of GFP+ pallial projection neurons are progenies of cells derived from the tectal region.

## Discussion

### Ontogeny of the zebrafish PG

By using the tamoxifen inducible Cre-lox system in zebrafish, we labeled with mCherry the cells that were located in the midbrain region during development (between 24 hpf and 6 wpf). The abundance of mCherry+ cells in the adult PG suggests that the majority of PG cells derive from the mesencephalic region. PG cells were consistently labeled with mCherry following the treatments at all the developmental stages examined. This suggests that PG progressively grows by addition of cells migrating from the mesencephalon.

It is difficult to prove whether all the PG cells originate from the mesencephalon, due to the technical limitation of the tamoxifen induction. Long term tamoxifen treatment leads to a high mortality rate of the fish during the experiment (*Bloch et al., 2019*; *Yu et al., 2020*). Moreover, the Cre-lox system would not allow 100% induction rate (*Hayashi and McMahon, 2002*). Thus, mCherry labeling of each experiment would represent only a fraction of the cells originating from the mesencephalic region.

With the use of the *Tg(279A-GFP)*, we could label visual pallial projection neurons in the PG (or more specifically in the PGl). There are only about 200 GFP+ neurons in the adult PG. In the quadruple transgenic lines expressing both GFP and mCherry, we observed consistently at least one or two co-expressing cells whenever the tamoxifen induction was performed before 6 wpf. Considering the short duration of each tamoxifen treatment, most of the GFP+ cells may originate from the midbrain. Thus, we conclude that the PGl projection neurons are mainly composed of cells migrating from the mesencephalon, by gradual accumulation from embryonic to juvenile stages (until around 6 wpf). The absence of mCherry/GFP co-localization at 8 wpf suggests that the PGl reaches maturation around this stage, as indicated by the abundance of projections to the Dl (*Figure 3C,D*).

At early embryonic stages (before 30 hpf) in both quadruple transgenic lines, the Cre expression appears to extend from dorsal to ventral MHB, therefore it is hard to conclude whether the PG cells derive from the alar or basal portion. A recent publication using *shh*-GFP transgenic line suggests that the adult PG contains *shh*-expressing cells (GFP+ in the transgenic line), indicating that some PG cells may be of

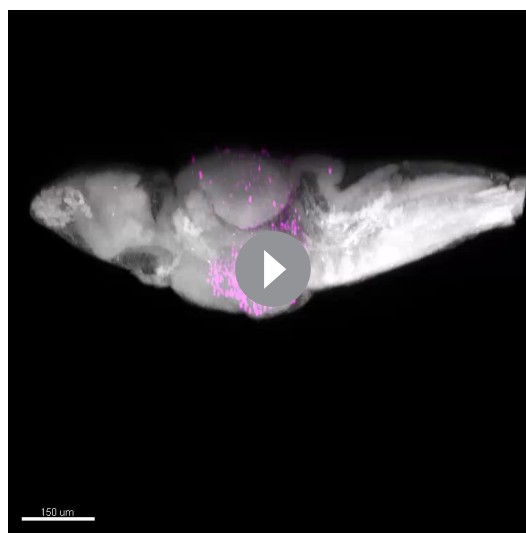

**Video 2.** 3D reconstruction of the mCherry+ cells in the 2 wpf brain of *Tg(her5:ERT2CreERT2;βactin:lox-stop-lox-hmgb1-mCherry)* zebrafish treated with tamoxifen at 24 hpf. mCherry+ cells are shown in magenta, and DiD fiber labeling is shown in grey.
https://elifesciences.org/articles/54945#video2

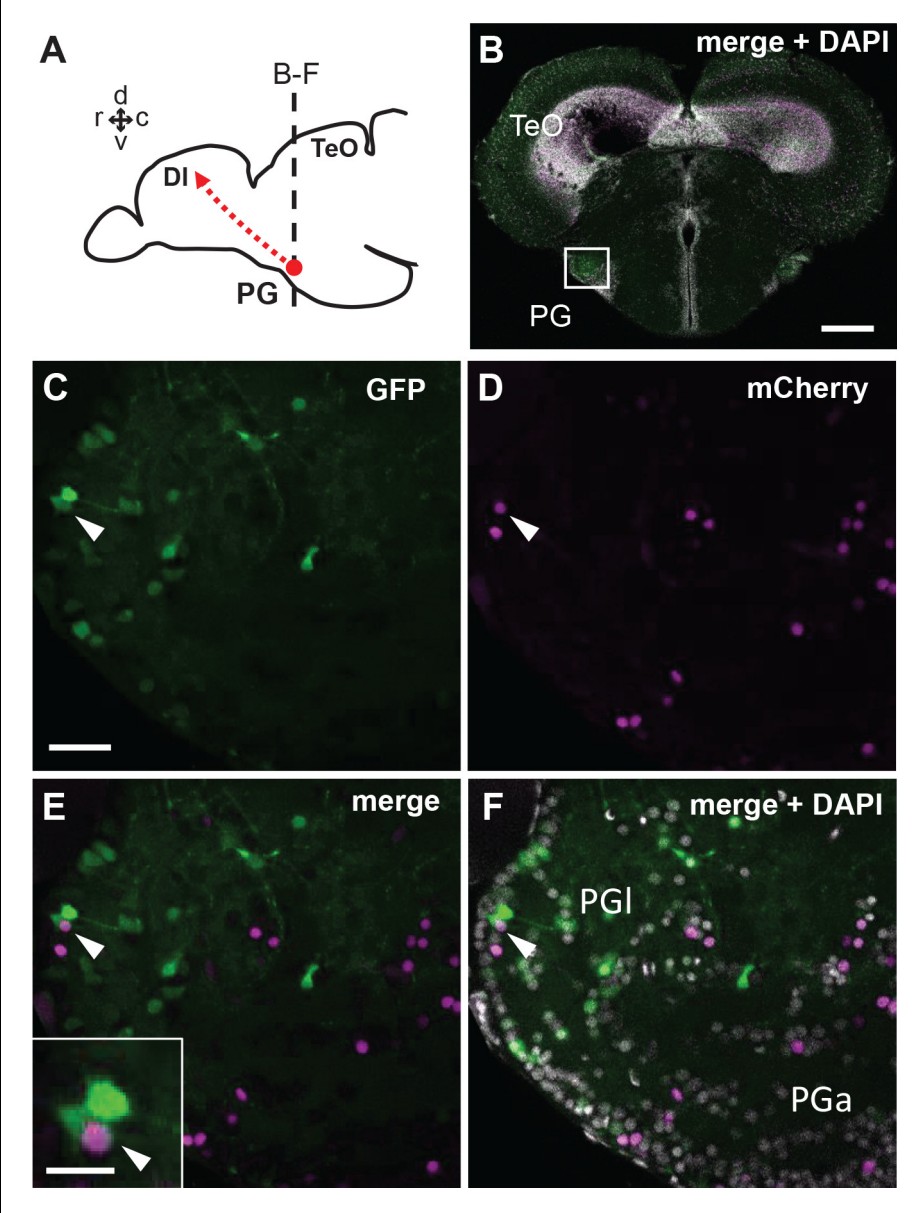

**Figure 6.** Co-localization of GFP and mCherry in the adult PG cells of the quadruple transgenic line *Tg(her5: ERT2CreERT2;βactin:lox-stop-lox-hmgb1-mCherry;279A-GFP)* following the tamoxifen induction at 24 hpf. (**A**) Schematic drawing of the adult zebrafish brain indicating the level of section through the PG shown in B-F. (**B**) A single plane confocal image showing a global view of the frontal section of a 3 mpf zebrafish brain. The white square indicates the PGl shown in C-F at a higher magnification. (**C–F**) A confocal image (5 µm projection) showing the co-localization of GFP and mCherry in PGl (arrowheads). Inset of **E** shows the double-labeled cell at a higher magnification. Abbreviations: Dl, lateral part of dorsal telencephalic area; PG, preglomerular complex; PGa, anterior preglomerular nucleus; PGl, lateral preglomerular nucleus; TeO, optic tectum. Brain orientation: r, rostral; c, caudal; d, dorsal; v, ventral. Scale bars = 200 µm (**B**); 30 µm (**C**; applicable to D-F); 10 µm (inset of E).
The online version of this article includes the following figure supplement(s) for figure 6:

**Figure supplement 1.** A simplified schema of the constructs of transgenic lines and order of crossing to generate the first quadruple transgenic fish: *Tg(her5:ERT2CreERT2;βactin:lox-stop-lox-hmgb1-mCherry;279A-GFP)*.

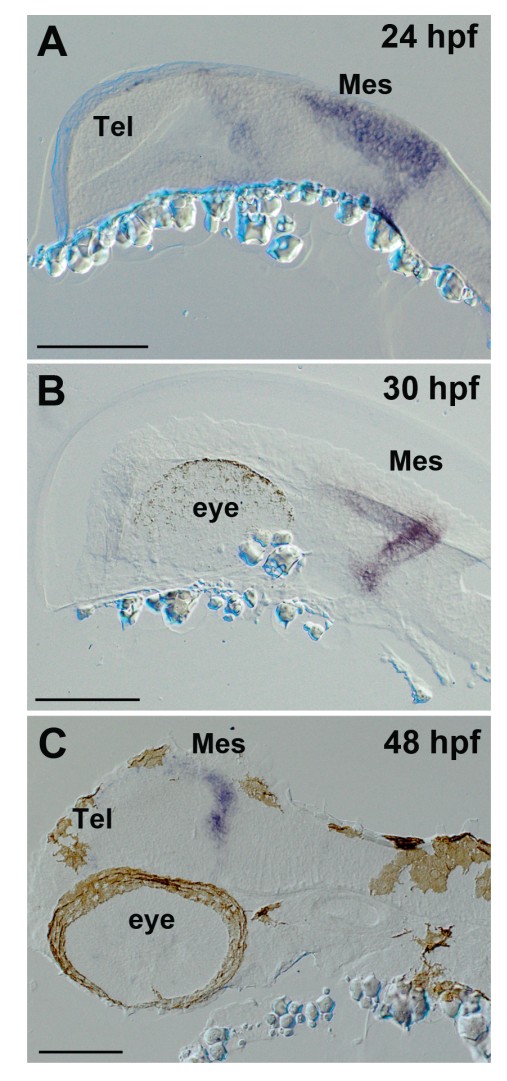

**Figure 7.** Expression of *ert2Cre* in the *Tg(Dr830: ERT2CreERT2)* embryonic brains. (**A–C**) Sagittal sections of embryonic brains (rostral to the left). At 24 hpf (**A**), *ert2Cre* is highly expressed in the mesencephalic domain, and a weak expression is also found in the anterior part of the brain. Later, at 30 hpf (**B**), the *ertCre* expression becomes limited to the mesencephalon. At 48 hpf (**C**), the *ertCre* expression is found exclusively in the tectal area. In C, the section plane is slightly tilted showing a more dorsal view of the embryo. Abbreviations: Mes, mesencephalic area; Tel, telencephalic area. Scale bars = 100 μm.

The online version of this article includes the following figure supplement(s) for figure 7:

**Figure supplement 1.** Expression of *ert2Cre* in the *Tg (Dr830:ERT2CreERT2)* juvenile brains.

basal origin (**Wullimann and Umeasalugo, 2020**). In contrast, the expression of Cre at later stages (after 48 hpf) is limited to the alar portion of the mesencephalon (notably tectal area). Thus, PG cells are migrating from the alar mesencephalic region at least after larval stages. This is consistent with previous studies claiming the PG is comprised of Pax6-expressing migrating cells (**Wullimann and Rink, 2001**; **Ishikawa et al., 2007**): both thalamic and tectal regions are Pax6 + domains.

This earlier data had been interpreted as demonstrating a forebrain origin of PG cells, implying that the Pax6+ cells are thalamic (**Ishikawa et al., 2007**), or prethalamic **Wullimann and Rink, 2001**; **Wullimann and Rink, 2002** following the prosomeric model (**Wullimann and Puelles, 1999**). The main argument of the forebrain origin of the PG is an indication of radial organization from the diencephalic ventricular zone to the putative PG region (**Wullimann and Rink, 2002**; **Ishikawa et al., 2007**; **Mueller and Guo, 2009**). In our data, trajectory of the migration was not clearly demonstrated, and we cannot exclude that some PG cells are radially migrated from the diencephalic ventricular zone. Thus, we cannot rule out the possibility that some radially migrated cells serve as a scaffold at an early stage, and mesencephalic cells by tangential migration may accumulate on them.

GFP+ fiber labeling in Dl appears to be prominent only at around 8 wpf (**Figure 3C**). Unlike the thalamocortical projections in amniotes that are already abundant at late embryonic stages (**Cordery and Molnár, 1999**; **Bielle et al., 2011**; **Molnár et al., 2012**), the visual ascending projections to the pallium in zebrafish are not mature until late juvenile stages. Yet zebrafish larvae can coordinate body orientation against the current, capture food, or escape from predators using relatively simple tectal circuitries (retina → TeO → motor outputs) (**Del Bene et al., 2010**; **Grama and Engert, 2012**; **Barker and Baier, 2013**). Such visuo-motor processing at the level of tectum (without reaching the forebrain) may be comparable to the circuitry involved in saccadic eye movements in mammals (**Yamamoto and Bloch, 2017**). It is possible that larval and early juvenile zebrafish behaviors are largely dependent on the tectal circuitries, and the visual system involving telencephalic circuitry becomes more important at later stages.

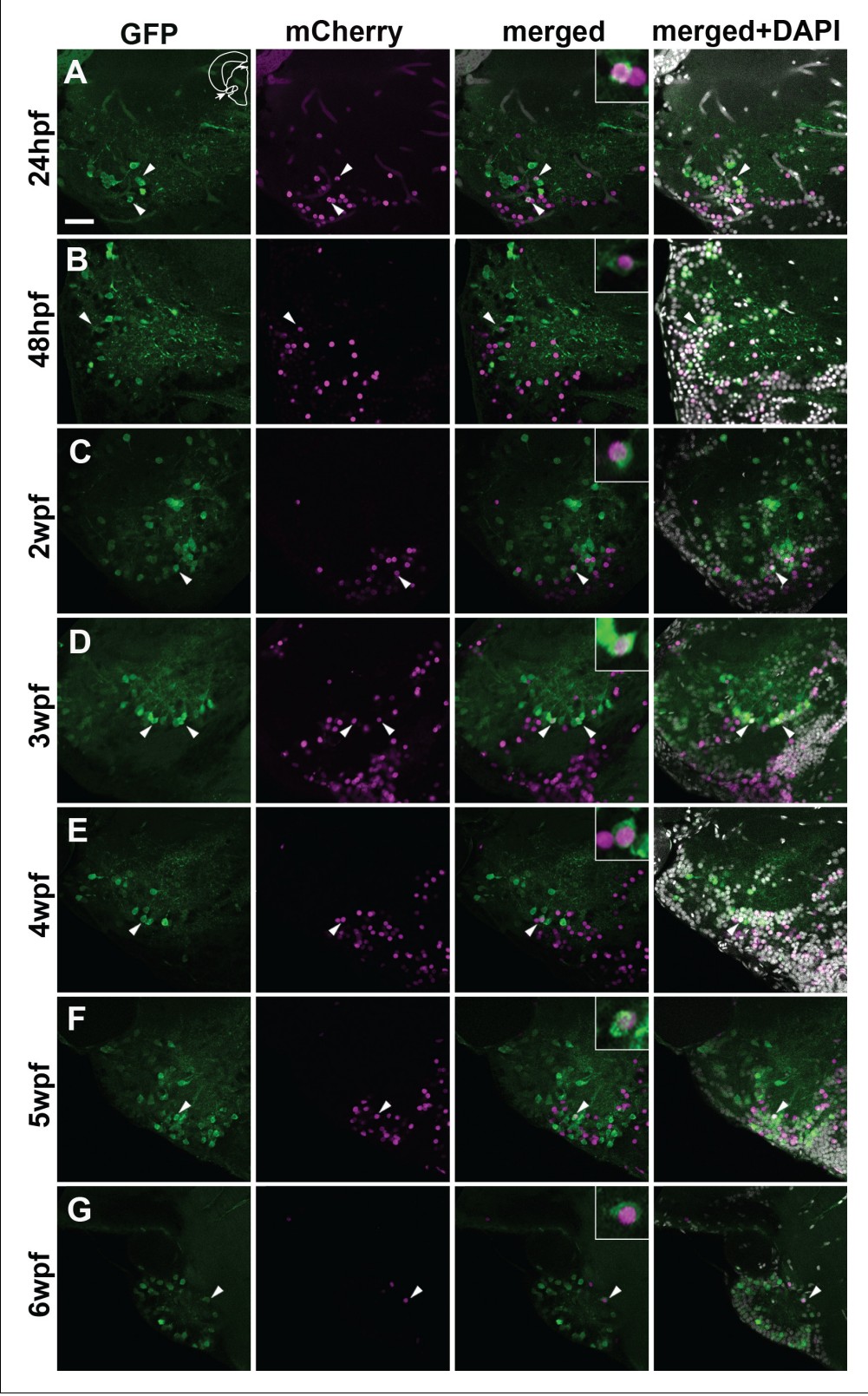

**Figure 8.** Co-localization of GFP and mCherry in the adult PG cells of the quadruple transgenic line *Tg(Dr830: ERT2CreERT2;βactin:lox-stop-lox-hmgb1-mCherry;279A-GFP)* following the tamoxifen induction at different developmental stages. (**A–G**) Confocal images (5 μm projection) of frontal sections through the PG showing the co-localization of GFP and mCherry (arrowheads). The GFP/mCherry co-localization was consistently observed in *Figure 8 continued on next page*

*Figure 8 continued*
the animals induced up to 6 wpf. Co-labelled cells are shown at a higher magnification in the insets in merged images. The section plane shown here is identical to that in *Figure 6*. Scale bar = 30 μm.
The online version of this article includes the following figure supplement(s) for figure 8:

**Figure supplement 1.** A simplified schema of the constructs of transgenic lines and order of crossing to generate the second quadruple transgenic fish: *Tg(Dr830:ERT2CreERT2;βactin:lox-stop-lox-hmgb1-mCherry;279A-GFP)*.
**Figure supplement 2.** Proportion of mCherry+ cells during development in the PG of the quadruple transgenic line *Tg(Dr830:ERT2CreERT2;βactin:lox-stop-lox-hmgb1-mCherry;279A-GFP)*.

## Evolution of ascending visual pathways

Based on mammalian studies, the presence of 'two visual systems' terminating in the mammalian cortex has been proposed (*Schneider, 1969*). One system is called the 'thalamofugal' or 'geniculate' pathway, in which retinal inputs reach the striate visual cortex (V1) via a thalamic nucleus (lateral geniculate nucleus in the case of mammals). The other is called the 'tectofugal' or 'extrageniculate' pathway, in which retinal inputs reach the extrastriate visual cortex via two intermediate structures, the tectum (superior colliculus in mammals) and another thalamic nucleus (pulvinar in primates and lateral posterior nucleus in other mammals like rodents). Since then, most studies in non-mammals have been interpreted based on this notion of 'two visual systems' (*Hall and Ebner, 1970*; *Karten and Hodos, 1970*; *Riss and Jakway, 1970*; *Ingle, 1973*; *Karten et al., 1973*; *Hagio et al., 2018*).

However, researchers have never reached a consensus on the evolutionary history of the tectofugal pathways. The debates have often been related to unsolved homology of the target pallial areas. The main issue is which part of the pallium would correspond to the mammalian neocortex in non-mammals such as birds (*Karten and Shimizu, 1989*; *Bruce and Neary, 1995*; *Striedter, 1997*; *Puelles et al., 2000*; *Butler et al., 2011*; *Dugas-Ford et al., 2012*) and teleosts (*Braford, 1995*; *Wullimann and Mueller, 2004*; *Northcutt, 2006*; *Yamamoto et al., 2007*; *Mueller et al., 2011*).

Our study does not solve the problem of pallial homology, but it strongly suggests that the tectofugal pathways in amniotes and teleosts are not homologous. In tetrapods, the neuronal cell bodies giving rise to the pallial projections are located in the thalamus, even though their projection targets differ between the major tetrapod groups. In contrast, our data clearly suggest that many (if not all) of the PG neurons are not part of the thalamus, but have a mesencephalic origin. Thus, in terms of regional homology, the teleost PG is not homologous to the amniote thalamus that is considered to be of forebrain origin. There exists a possibility that some PG cells come from a forebrain territory, as suggested by previous studies (*Wullimann and Rink, 2001*; *Wullimann and Rink, 2002*; *Ishikawa et al., 2007*). Nonetheless, in addition to the developmental criteria, the cladistics analysis also reveals the lack of evolutionary continuity of the thalamo-pallial projections as detailed below.

Substantial tectofugal visual pathways to the pallium are observed only in amniotes and teleosts, and not in the intermediate taxa (*Figure 9*). In amphibians, the major sensory relay nucleus is the dorsal thalamus, but unlike the amniote situation, there are very few projections to the pallium, and the majority terminates in the subpallium (ventral telencephalon) (*Kicliter, 1979*; *Neary and Northcutt, 1983*; *Wilczynski and Northcutt, 1983*; *Butler, 1994a*). Similarly, basal groups of the actinopterygians also have a poorly developed pallial connectivity, and afferent projections from the thalamic region seem to terminate in the ventral portion of the telencephalon (*Albert et al., 1999*; *Yamamoto et al., 1999*; *Holmes and Northcutt, 2003*). Furthermore, in *Polypterus*, visual projection to the pallium is mediated via the nucleus medianus of the posterior tuberculum (MTP; *Figure 9*), which is considered to be uniquely derived in this group and not homologous to any known pathways in tetrapods (*Northcutt et al., 2004*; *Northcutt, 2009*). Indeed, there are more than two ascending pathways, with a variable abundance of each pathway depending on species (*Riss and Jakway, 1970*; *Graybiel, 1972*; *Benevento and Standage, 1983*; *Gamlin and Cohen, 1986*; *Albert et al., 1999*; *Wild and Gaede, 2016*; *Heap et al., 2017*).

In combination, these observations strongly suggest that the teleost and amniote pathways are not homologous to one another, but have evolved independently.

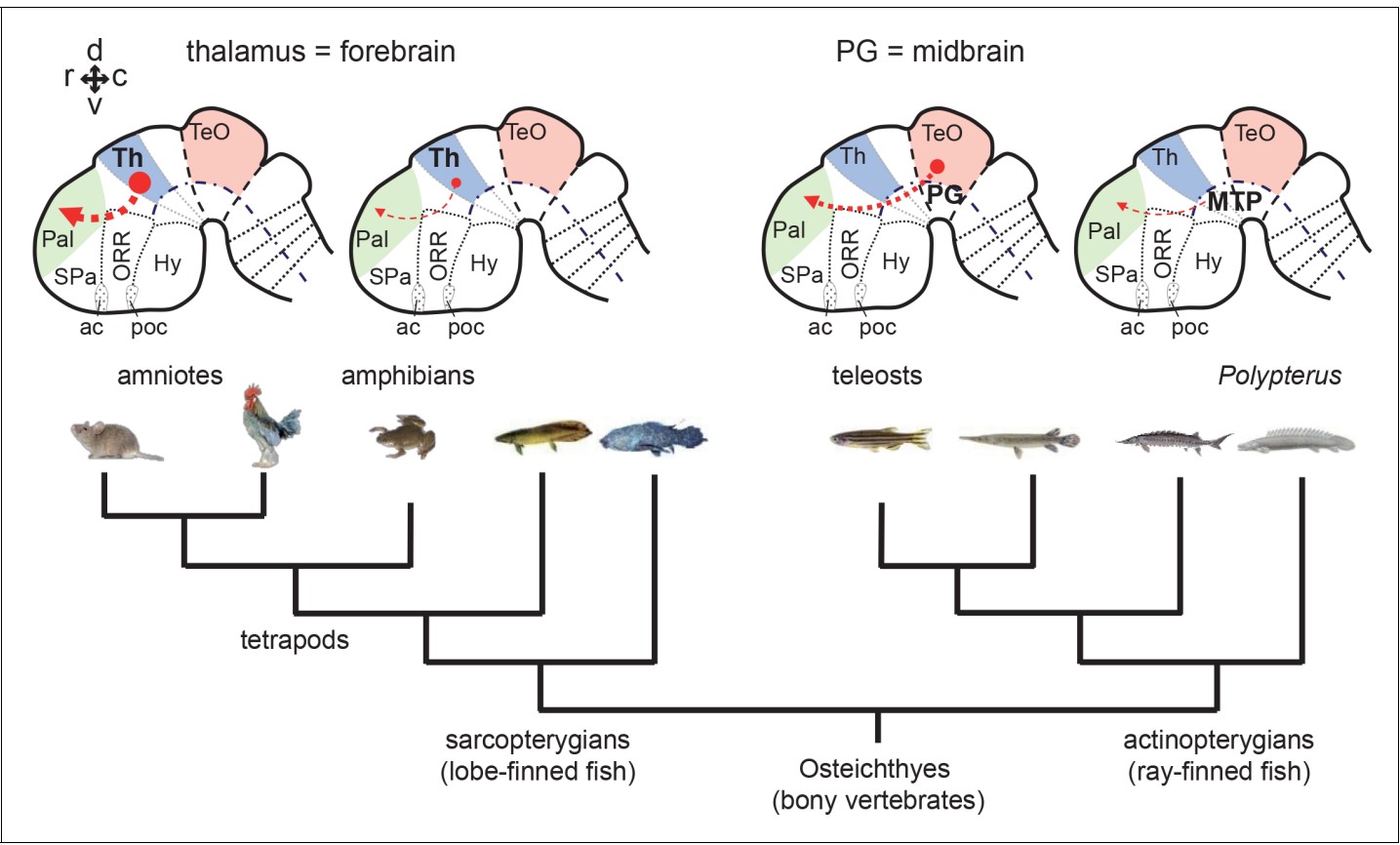

**Figure 9.** Phylogenetic comparison of pallial projections in bony vertebrates (Osteichthyes). Schematic drawing of representative embryonic brains of sarcopterygians (amniote and amphibian) and actinopterygians (teleost and *Polypterus*) are shown above the phylogenetic tree of the *Osteichthyes*. Afferent projections to the pallium are shown with red arrows. The thickness of the line represents the relative prominence of the pallial projections in each group. As suggested in the present study, cell populations giving rise to pallial projections are unlikely derived from the thalamic region in actinopterygians. The basal groups of sarcopterygians and actinopterygians (e.g. amphibians and *Polypterus*) have poorly developed pallial projections. Therefore, abundant pallial projections are likely to have evolved independently in amniote and teleost lineages. Abbreviations: ac, anterior commissure; Hy, hypothalamus; MTP, nucleus medianus of the posterior tuberculum; ORR, optic recess region; Pal, pallium; PG, preglomerular complex; poc, post-optic commissure; SPa, subpallium; TeO, optic tectum; Th, thalamus. Brain orientation: r, rostral; c, caudal; d, dorsal; v, ventral.

### Thalamocortical-like functions by midbrain neurons in teleosts?

The amniote thalamo-pallial projections (thalamocortical projections in mammals) are intra-forebrain projections, from the dorsal diencephalon to the dorsal telencephalon. Due to the enlargement of the forebrain in mammals especially in humans, forebrain evolution has drawn much attention for the study of sensory processing and cognitive functions.

In contrast, our data suggest that the PG-pallial projection in teleosts is a midbrain-forebrain projection, and that a non-forebrain cell population can play an equivalent role to the thalamocortical projection neurons (conveying sensory information to the pallium). We here focused on the tectofugal visual pathway through the PGl because of the availability of a zebrafish transgenic line, but the PG as a whole also receives inputs of other sensory modalities (auditory, lateral line, gustatory etc) (*Striedter, 1991*; *Striedter, 1992*; *Yoshimoto et al., 1998*; *Yamamoto and Ito, 2005*; *Yamamoto and Ito, 2008*; *Northcutt, 2006*). We have previously shown that the external part of the IL, another teleost-specific multi-sensory integration center (*Rink and Wullimann, 1998*; *Shimizu et al., 1999*; *Ahrens and Wullimann, 2002*; *Yang et al., 2007*), is also derived from the mesencephalon (*Bloch et al., 2019*). More studies are needed to determine the functions of PG and IL, but these data indicate that the teleost lineage have taken an evolutionary path different from amniotes, recruiting more mesencephalic structures for sensory processing.

In addition to the variation in neuronal connectivity that we reveal here, we have already demonstrated unexpected diversity of dopamine systems, despite their involvement in similar physiological/behavioral properties across vertebrate groups (*Fontaine et al., 2015*; *Yamamoto et al., 2015*; *Yamamoto et al., 2017*; *Yamamoto and Bloch, 2017*). Recently, *Striedter and Northcutt, 2020* have also pointed out a number of examples of convergent evolution across vertebrate taxa, in agreement with our hypothesis. Taking all these data into consideration, vertebrate brains have tremendously diversified across taxa during evolution, and many more similarities are due to convergent evolution than previously thought.

## Materials and methods

### Zebrafish lines

For tract-tracing study, wild-type zebrafish (*Danio rerio*) with the Oregon AB genetic background of both sexes were used.

The *Tg(gSAGFF279A)* and *Tg(UAS:GFP)* transgenic lines were generated in the National Institute of Genetics (Mishima, Japan) (*Asakawa and Kawakami, 2009*; *Kawakami et al., 2010*; *Lal et al., 2018*), and their offspring *Tg(gSAGFF279A;UAS:GFP)*, abbreviated as *Tg(279A-GFP)*, were used in this study. This zebrafish line was maintained either by incross or by crossing with AB.

The *Tg(279A-GFP)* fish line was crossed with other transgenic lines to perform cell lineage studies. It was crossed either with the *Tg(her5:ERT2CreERT2)* (*Galant et al., 2016*) or *Tg(Dr830:ERT2-CreERT2)* (*Heuzé, 2017*), plus with the *Tg(βactin:lox-stop-lox-hmgb1-mCherry)* (*Wang et al., 2011*; *Galant et al., 2016*), thus obtaining the quadruple *Tg(her5:ERT2CreERT2;βactin:lox-stop-lox-hmgb1-mCherry;279A-GFP)* or *Tg(Dr830:ERT2CreERT2;βactin:lox-stop-lox-hmgb1-mCherry;279A-GFP)*.

For in situ hybridization (ISH) of *ert2Cre*, double transgenic lines *Tg(her5:ERT2CreERT2;βactin:lox-stop-lox-hmgb1-mCherry)* and *Tg(Dr830:ERT2CreERT2;βactin:lox-stop-lox-hmgb1-mCherry)* were used.

### Fish maintenance and staging

Zebrafish used for the tract tracing were maintained at Nagoya University (Japan) in aquaria at 22–26°C. For the rest of the experiments, zebrafish were raised in the animal facility in Neuro-PSI (Gif-sur-Yvette, France). Embryos/larvae up to 5 days post-fertilization (dpf) were maintained and staged as described (*Kimmel et al., 1995*). After larval stages, zebrafish were raised in a fish facility (maintained at 26–28°C). Zebrafish at 3 mpf or older is considered as adult. In all experiments performed in this study, we randomly used both male and female.

The experimental protocols and care of laboratory animals were conducted in compliance with the official regulatory standards and approval of the French Government (reference document n° APAFIS#1286–2015062616102603 v5), the official Japanese regulations for research on animal, and the regulations on Animal Experiments in Nagoya University.

### DiI tract-tracing

To examine neural connectivity in the adult zebrafish, we placed crystals of DiI (1,1′-dilinoleyl-3,3,3′,3′-tetramethylindocarbocyanine, 4-Chlorobenzenesulfonate; FAST DiI solid, Thermo Fisher Scientific-Molecular Probes, D7756) in the telencephalon and the optic tectum *post-mortem*. DiI is a fluorescent lipophilic tracer that diffuses along lipid membranes, allowing both anterograde and retrograde labeling of neural processes. Adult zebrafish (n = 40) were fixed with 4% paraformaldehyde (PFA) in phosphate buffered saline (PBS) overnight at 4°C.

The brains were dissected out and a small crystal was inserted into the brain using a glass pipette. The crystal was left in the brain for dye diffusion from 10 days to 2 weeks at 37°C in PBS, or PBS containing 0.05% sodium azide to avoid fungal contamination. The brains were embedded in 3% agarose, and sectioned at 80 μm (in frontal and sagittal) using a vibratome (Leica VT 1000 s).

### Biocytin and BDA tract-tracing

Biocytin (Sigma-Aldrich, B4261) was injected into adult zebrafish brains (n = 40), both in vivo and in vitro. For some in vivo injections into the TeO, BDA (molecular weight 3000; Thermo Fisher

Scientific-Molecular Probes, D7135) was also used (n = 3), because labeled terminals in the PGl were clearer than those with biocytin.

For in vivo tract-tracing, fish were anesthetized by immersing in fresh water containing 200 mg/L tricaine methanesulfonate (MS222; Sigma-Aldrich, A5040) and set in a device for physical restraint. A small amount of fresh water containing 150–200 mg/L MS222 were poured on the fish for aeration and also to maintain the anesthetic condition. Prior to injections of biocytin into the optic nerve, extraocular muscles and the optic nerve were cut, and the eyeball was excised. Crystals of biocytin were injected into the proximal stamp of the optic nerve with a minute insect pin. For injections of biocytin into TeO and Dl, a dorsal portion of the cranium was opened with forceps to expose the brain, and crystals of biocytin were inserted into the target region with a minute insect pin.

For BDA injections into the TeO, the fish were aerated with fresh water containing 85 mg/L MS222 through the mouth to maintain the anesthetic condition. A dorsal portion of the cranium was opened with a dental drill (Minimo ACE; MNA Minitor Co. Ltd.) to expose the brain, and a glass microelectrode (tip diameters: 4–14 µm) filled with 1–2% BDA solution in 0.05M Tris-HCl-buffered saline (TBS; pH 7.4) was driven into the TeO with a manipulator (MN-3; Narishige). BDA was injected iontophoretically with square current pulses (+5 µA, 0.5 Hz, 50% duty cycle) passed through the electrodes for 5 min with a stimulator (SEN-3301; Nihon Kohden). After the injection, the orbital cavity and the cranial opening were closed with a flap made of a paraffin sheet (Parafilm, Bemis Company) or a small piece of Saran Wrap; both were affixed to the cranium with an acrylic adhesive (Aron alpha, jelly type; Toagosei).

Postoperative fish were maintained in aquaria for 1–5 hr. After the survival period, the fish were deeply anesthetized with MS222 (over 200 mg/L) and perfused through the heart with 2% paraformaldehyde and 1% glutaraldehyde in 0.1 M phosphate buffer (PB), pH 7.4. The brains were removed from the skull and post-fixed in fresh solution of the same fixative at 4°C for 1 to 2 days.

We also injected biocytin into the TeO, PGl, and Dl in vitro because it was difficult to maintain postoperative fish in aquaria for hours following injections in vivo. A detailed in vitro tract-tracing method has been reported previously (*Yamamoto and Ito, 2008*). Fish were deeply anesthetized with MS222 (over 200 mg/L). We quickly dissected the brain from the skull and then injected crystals of biocytin into TeO, PGl, and Dl with a minute insect pin. The brain was kept in a container filled with 50 mL normal artificial cerebrospinal fluid solution for marine teleosts (126 mM NaCl, 4.0 mM KCl, 1.0 mM $MgSO_4$, 1.7 mM $CaCl_2$, 26 mM $NaHCO_3$, 1.0 mM $NaH_2PO_4$, and 10 mM glucose; *Tsutsui et al., 2001*) at room temperature. The solution was aerated and changed every 30 min. After 3–4.5 hr, we fixed the brain by immersion in 2% paraformaldehyde and 1% glutaraldehyde in 0.1 M PB for 1–3 days at 4°C.

## Tissue processing following biocytin and BDA injections

The fixed brains were cryo-protected by immersion in 0.1 M PB containing 20% sucrose at 4°C overnight. Cryo-protected brains were embedded in 5% agarose (type IX, ultra-low gelling temperature; Sigma-Aldrich, A2576) containing 20% sucrose and frozen in n-hexane at −60°C. Then, frontal sections were cut at a thickness of 40 µm on a cryostat and mounted on gelatin-coated glass slides. The sections were dried for one hour at room temperature and washed once with 0.05 M TBS containing 0.1% Tween 20 (TBST) and twice with TBS each for 10 min. To quench non-specific peroxidase activities, sections were steeped in methanol containing 0.3% $H_2O_2$ for 10 min and washed three times with TBS and once with 0.03% TBST each for 10 min. Sections were incubated with a solution of avidin-biotin-peroxidase complex (1:100; VECTASTAIN Elite ABC Standard Kit, Vector Laboratories, PK-6100) overnight at room temperature. After a wash with TBST and three washes with TBS each for 10 min, sections were incubated for one hour with 0.05% 3,3'-diaminobenzidine (Sigma-Aldrich, D5637) solution in 0.1 M PB containing 0.04% nickel ammonium sulfate and 0.01% $H_2O_2$. The reaction was stopped by four times washes with TBS, and the sections were counterstained with 0.05–0.1% cresyl violet, dehydrated, and coverslipped with Permount (Fisher Scientific, SP15-500).

One fish was used for histological analysis with Nissl-stained sections (frontal section). Fixation and tissue processing were performed as mentioned above for in vivo tract-tracing materials, except that the section thickness was 30 µm. After going through an ascending series of ethanol to remove lipid, the sections were rehydrated through a descending series of ethanol. The sections were stained with 0.1% cresyl violet, dehydrated, cleared in xylene, and coverslipped with Permount.

## Tamoxifen treatment

Tamoxifen treatments were performed in quadruple transgenic fish (see above) as described previously (*Galant et al., 2016*; *Bloch et al., 2019*). 4-Hydroxytamoxifen (Sigma-Aldrich, T176) was dissolved in ethanol at a concentration of 10 mg/ml and stored at −20°C until use. The working solution was freshly prepared before the treatment, then further diluted with embryo medium (for 24 hpf, 30 hpf, and seven dpf) or fish water (for 2–6 wpf). The fish were incubated in the tamoxifen working solution at 28°C in the dark.

Embryos at 24 hpf and 30 hpf were dechorionated with Pronase (1 mg/ml; Sigma-Aldrich, P5147) prior to the tamoxifen treatment. Embryos were placed into the six-well culture plate (Thermo Fisher Scientific) and were incubated with embryo medium containing tamoxifen. 24 hpf embryos were treated with 10 µg/ml tamoxifen for 6 hr, and 30 hpf embryos were treated with 5 µg/ml tamoxifen for 24 hr. After the incubation, the fish were washed rapidly three times with embryo medium, then put back to the incubator. seven dpf larvae were treated in a large petri dish (around 100 ml embryo medium) with 5 µg/ml tamoxifen on two consecutive days, with an incubation time of 4 hr each.

For juveniles (2–6 wpf), fish were placed in a beaker (100–200 ml fish water depending on the number of fish) with an air pump, and incubated with 2 µg/ml tamoxifen on four consecutive days. The incubation time per day was 2–4 hr, and the treatment was interrupted whenever the fish looked sick. At the end of each incubation, the fish were gently washed 3 times with fish water, placed back to a clean fish tank and fed.

The tamoxifen treatments were performed at least twice per each developmental stage, and each treatment contained at least 10 individuals (see *Supplementary file 1* for number of fish analyzed). The tamoxifen-induced mCherry expression was systematically observed at 3 mpf. The fish were sacrificed and double-immunofluorescence anti-GFP and anti-dsRed were performed (see below).

## Tissue preparations for immunofluorescence or in situ hybridization (ISH)

Zebrafish embryos up to 48 hpf were fixed in ice-cold 4% paraformaldehyde (PFA; Electron Microscopy Sciences) in 0.01 M PBS containing 0.1% Tween 20 (PBST) overnight at 4°C. Zebrafish older than 5 dpf were deeply anesthetized using 0.2% tricaine methanesulfonate (MS222; Sigma-Aldrich) diluted in fish water. The fish were fixed in 4% PFA in PBST overnight at 4°C, then brains were dissected out.

Samples used for ISH were dehydrated in ethanol gradient series, and kept at −20°C in methanol at least for a couple of days. They were rehydrated prior to ISH. For immunolabeling, samples were conserved in a stocking solution containing 0.5% PFA and 0.025% sodium azide. Adult brains were sectioned in a frontal plane (80 µm) with a vibratome.

For a whole-brain imaging for zebrafish younger than 2 wpf (14 dpf), a simplified clearing protocol was applied as previously described (*Affaticati et al., 2018*; *Bloch et al., 2019*). Depigmentation was applied as follows: up to 15 larvae were incubated in 10 mL of pre-incubation solution in a petri dish (0.5X saline sodium citrate buffer (SSC), 0.1% Tween20) for 1 hr at room temperature without stirring. Then samples were bleached by incubation in depigmentation solution (0.5X SSC, 5% formamide, 3% $H_2O_2$). Samples were left in the solution until pigments were completely degraded. Samples were then washed three times in PBST and left overnight in PBST.

For tissue clearing of older zebrafish brains, a passive CLARITY technique (zPACT) was performed (*Affaticati et al., 2017*). Dissected brains were fixed in freshly prepared ice-cold methanol-free 4% PFA in PBS (pH 7.4) at 4°C overnight. Samples were then soaked in a precooled solution of hydrogel (0.01 M PBS, 0.25% VA-044 initiator, 5% dimethyl sulfoxide, 1% PFA, 4% acrylamide, and 0.0025% bis-acrylamide) at 4°C for 2 days. The hydrogel polymerization was triggered by replacing atmospheric oxygen with nitrogen in a desiccation chamber at 37°C for 3 hr. Passive tissue clearing was performed at 37°C for 5 days in the clearing solution (8% SDS, 0.2 M boric acid, pH adjusted to 8.5) under rotation in a hybridization oven. After clearing, brains were washed in PBST at room temperature with gentle shaking for 2 days. Brains were incubated in a depigmentation solution (0.5X SSC (150 mM NaCl, 15 mM sodium citrate, pH 7.2), 5% formamide, 0.5X SSC, 3% $H_2O_2$, 0.1% Tween 20) for 40 min under light until all remaining pigments were bleached. After washing in PBST brains were post-fixed in PFA 4% in PBS (pH 7.4) at 4°C overnight.

## Immunofluorescence

Double immunolabeling for GFP (1:1000; Aves Labs, GFP-1020; RRID:AB_10000240) and dsRed (1:600, Takara Bio, 632496; RRID:AB_10013483) was performed on adult brain sections of quadruple transgenic zebrafish. Samples were incubated in primary antibodies in PBST containing 4% NGS and 0.3% Triton X-100 at 4°C overnight. Then, samples were incubated with secondary antibodies conjugated to fluorophores (1:1000; Alexa Fluor 488 and 546, Thermo Fisher Scientific-Molecular Probes) in PBST at 4°C overnight. Goat anti-chicken antibody labelled with Alexa Fluor 488 (A-11039; RRID: AB_142924) was used for anti-GFP, and goat anti-rabbit antibody labelled with Alexa Fluor 546 (A-11010; RRID:AB_143156) was used for dsRed. The same protocol was used for single-color immunofluorescence (either GFP or dsRed only) on adult and juvenile brain sections of the parent transgenic lines. In order to visualize the brain morphology, the sections were counterstained with DAPI (4′,6-diamidino-2-phenylindole dihydrochloride; 5 µg/ml, Sigma-Aldrich) at room temperature for 20 min.

Immunofluorescence on larval zebrafish brains *in toto* (14 dpf or earlier) was performed as described previously (*Bloch et al., 2019*). Briefly, after depigmentation, samples were incubated at room temperature for 5 hr in a blocking solution containing NGS, 10% dimethyl sulfoxide (DMSO), 5% PBS-glycine 1 M, 0.5% Triton X-100, 0.1% deoxycholate, and 0.1% NP-40 in PBST. Samples were then incubated in labeling solution (2% NGS, 20% DMSO, 0.05% sodium azide, 0.2% Triton X-100, PBST, 10 µg/ml heparin) with anti-dsRed antibody (1:600) at room temperature for 3–4 days with gentle shaking on a 3D rocker. Secondary antibody incubation was performed for 3–4 days and DiD (Thermo Fisher Scientific-Molecular Probes L7781; 1 µg/ml) labeling was added from the second day to show the whole brain morphology. Finally, samples were incubated in a fructose-based high-refractive index (RI) solution that is adjusted to 1.457 for imaging.

CLARITY-processed adult brains were incubated in blocking solution (0.01 M PBS, 0.1% Tween 20, 1% Triton X-100, 10% dimethyl sulfoxide, 10% normal goat serum, 0.05 M glycine) at 20°C for 3 hr. Subsequently samples were incubated in labeling solution (0.01 M PBS, 0.1% Tween 20, 0.1% Triton X-100, 9% dimethyl sulfoxide, 2% normal goat serum, 0.05% sodium azide) with the chicken anti-GFP antibody (1:400) for 7 days at room temperature under gentle agitation. After four washing steps in PBST, samples were incubated in labeling solution with the goat anti-chicken antibody labelled with Alexa Fluor 488 (1:400) at room temperature for 7 days. Samples were washed for 2 days in PBST and mounted in a fructose-based high refractive index solution (fHRI); 70% fructose, 20% DMSO in 0.002 M PBS, 0.005% sodium azide. The refractive index of the solution was adjusted to 1.457 using a refractometer (Kruss). The clarified samples were incubated in 50% fHRI for 6 hr and further incubated in 100% fHRI for 1 day. For imaging, samples were mounted in 1% low melting point agarose and covered with fHRI.

## In situ hybridization (ISH)

ISH for *ert2Cre* (*Dirian et al., 2014*; *Galant et al., 2016*; *Heuzé, 2017*) was performed for zebrafish brains of different developmental stages, in order to verify the expression of Cre recombinase. Detailed ISH procedures have been described in our previous publications (*Affaticati et al., 2015*; *Xavier et al., 2017*).

After rehydration, the samples were permeabilized with proteinase K (1 µg/ml; Sigma-Aldrich, P6556) at 37°C for 5–10 min. The proteinase K reaction was stopped by incubation with 2 µg/µl glycine. After PBST washes, the samples were incubated in hybridization buffer at 65°C for 4 hr, then hybridized with 2 ng/ml of cRNA probe in hybridization buffer at 65°C for at least 18 hr. Samples were then washed in gradient series of formamide/2X SSC mixture at 65°C: 75% formamide/25% 2X SSC, 50% formamide/50% 2X SSC, 25% formamide/75% 2X SSC, then washed in 2X SSC and finally in 0.2X SSC. After being rinsed with PBST at room temperature, the samples were incubated with anti-digoxigenin conjugated with alkaline phosphatase (1:2500; sheep anti-DIG-AP Fab fragments, Roche Diagnostics, 11093274910; RRID:AB_514497) at 4°C overnight. After PBST washes, the signal was visualized by incubation with nitroblue tetrazolium chloride (NBT) and 5-bromo-4-chloro-3-indolylphosphate (BCIP) solution (Roche Diagnostics, 11681451001) in 0.1 M Tris-HCl (pH9.5)/0.1 M NaCl in $H_2O$ (TN buffer).

For embryos, the entire ISH procedures were performed *in toto*. Embryos were embedded into 3% agarose and sectioned with a vibratome in a sagittal plane (40 µm), and slide mounted for

imaging. For juvenile brains, probe hybridization was performed *in toto*, and the brains were sectioned with a vitratome in a frontal plane (40 µm) before incubating with anti-DIG-AP.

Each experimental condition contained at least 4 samples for juvenile brains and at least 10 samples for embryos. The experiments were repeated at least 3 times for each developmental stage of tamoxifen treatments (*Supplementary file 1*), except for 8 wpf, in which we did not observe any mCherry/GFP co-localization (see Results).

## Image acquisition

A Leica TCS SP8 laser scanning confocal microscope was used to image adult sections with a 25x or 40x water immersion objective. For clarified brains, the same microscope was used with a Leica HC Fluotar L 25x/1.00 IMM motCorr objective. For all these acquisitions, fluorescence signal was detected through photomultipliers (PMTs) after sequential laser excitation of fluorophores at 405, 488, 552 nm. Steps along the Z-axis were set at 1 µm. Epifluorescence images were acquired using a Multizoom AZ100 (Nikon).

Bright-field images were acquired with upright microscopes, either BX43 or BX60 (Olympus). Acquired images were adjusted for brightness and contrast using ImageJ/FIJI software (*Schindelin et al., 2012*).

## Quantification of mCherry+ cells in PG

The mCherry+ cells in the adult PG were counted from confocal images using the ImageJ cell counter plugin. We used stacks of 5 µm from frontal sections containing the PGl. The total number of cells was determined with DAPI nuclear labeling, and proportion of mCherry+ cells was calculated. The cell count was performed in the brains induced at 4 different time points: 24 hpf, 48 hpf, 2 wpf, and 3 wpf, and the average from two specimens was presented as data for each time point (*Figure 8—figure supplement 2*).

## 3D image reconstruction of young zebrafish brains

In order to visualize the global distribution of mCherry+ cells in the brains of larval/juvenile (5, 7, 14 dpf) zebrafish, 3D reconstruction of confocal images was performed as described in *Bloch et al., 2019*. A whole brain imaged with confocal microscopy was reconstructed in 3D, using Imaris 8.0.1 software (Oxford Instruments) by means of the '3D view' visualization tool on a Dell T3610 workstation.

## Selective visualization of PGl fiber projections in *Tg(279A-GFP)*

The signal of the *Tg(279A-GFP)* was selectively visualized by manual segmentation of the fiber projections in Amira (Thermo Fisher Scientific, FEI). For the classification of the staining of the specimen, we broke down the staining pattern into four categories: background, specimen background, specimen signal, and specimen auxiliary signal that contains a widely distributed population of cells outside the scope of this study. The process of manual segmentation is an iterative succession of initial freehand (Brush) segmentation and subsequent refinement with interactive thresholding tools (Magic wand, Threshold). The segmentation was initialized by a coarse manual segmentation of the pattern of interest and was refined region by region in up to seven iterations. The rather high number of manual and threshold-based segmentation iterations is due to the auxiliary signal, which is also located in close proximity to the pattern of interest. Local thresholds for the (negative) segmentation of the auxiliary signal were adapted with respect to the voxel values of the surrounding region to a lower boundary value of 150–180 (higher boundary always was 255). The resulting regions of exceptionally bright voxels were dilated in all three dimensions by two pixels for removing the auxiliary signal in their entirety from the region of interest.

For the final visualization we multiplied the original grey value dataset with the binary 3D mask of the segmentation. Since in this mask the region of interest is encoded as one (1) and the background as zero (0) the multiplication of the image with its mask will result in the separation of its original grey values within the mask from the background signal because the latter is multiplied by zero. This isolated signal was visualized using the orange-yellow colormap (look-up-table, LUT) volrenRed.col with a truncated dynamic range (0–200) for better contrast and visibility of the signal of interest in a Volren-module of Amira.

For the context of the signal of interest, we visualized in parallel the entire dataset in a separate Volren-module using a grey colormap with slightly truncated dynamic range (10-255) as a means of global contrast enhancement.

## Acknowledgements

The authors thank members of Kawakami's team for their contribution for generating *Tg (gSAGFF279A)* and *Tg(UAS:GFP)* zebrafish transgenic lines. Many thanks to the members of TEFOR, notably Elodie de Job, Laurie Rivière, Elodie Machado, and Isabelle Robineau for their technical help. We also thank Matthieu Simion for his help and discussion, and Catherine Pasqualini for critical reading. We also thank members of the animal facility (especially Krystel Saroul) for taking care of zebrafish. Finally, we thank members of Laure Bally-Cuif's, Jean-Stéphane Joly's, and Philippe Vernier's team, for their support including technical, administrative, and financial help.

## Additional information

### Funding

| Funder | Grant reference number | Author |
| --- | --- | --- |
| Agence Nationale de la Recherche | ANR-13-BSV4-0001 PALL-E-NODY | Kei Yamamoto |
| Fondation pour la Recherche Médicale | Equipes FRM 2017 | Kei Yamamoto |
| Fondation pour la Recherche Médicale | Fin de thèse de sciences FDT201805005408 | Solal Bloch |
| Japan Society for the Promotion of Science | JP16J03625 | Hanako Hagio |
| Japan Agency for Medical Research and Development | NBRP/Genome Information upgrading program | Koichi Kawakami |
| National Institut of Genetics and NBRP | NIG-JOINT (2013-A15) | Koichi Kawakami |

The funders had no role in study design, data collection and interpretation, or the decision to submit the work for publication.

### Author contributions

Solal Bloch, Conceptualization, Funding acquisition, Validation, Investigation, Visualization, Methodology, Writing - original draft, Writing - review and editing, Conducted cell lineage study; Hanako Hagio, Funding acquisition, Validation, Investigation, Visualization, Methodology, Writing - original draft, Writing - review and editing, Conducted tract-tracing study; Manon Thomas, Investigation, Methodology, Conducted cell lineage study; Aurélie Heuzé, Resources, Created the Tg(Dr830:ERT2-CreERT2) transgenic line; Jean-Michel Hermel, Investigation, Visualization, Writing - review and editing; Elodie Lasserre, Ingrid Colin, Investigation; Kimiko Saka, Resources, Visualization, Created and characterized the Tg(gSAGFF279A;UAS:GFP) transgenic line; Pierre Affaticati, Investigation, Methodology, Performed tissue clearing and whole-brain image acquisition; Arnim Jenett, Data curation, Formal analysis, Visualization, Methodology, Performed 3D reconstruction of confocal images; Koichi Kawakami, Resources, Supervision, Funding acquisition, Validation, Visualization, Methodology, Created and characterized the Tg(gSAGFF279A;UAS:GFP) transgenic line; Naoyuki Yamamoto, Supervision, Funding acquisition, Validation, Methodology, Writing - original draft, Writing - review and editing, Supervised tract-tracing study; Kei Yamamoto, Conceptualization, Supervision, Funding acquisition, Validation, Investigation, Visualization, Methodology, Writing - original draft, Project administration, Writing - review and editing

## Author ORCIDs
Hanako Hagio (iD) https://orcid.org/0000-0003-2197-4595
Jean-Michel Hermel (iD) https://orcid.org/0000-0001-7902-8994
Kimiko Saka (iD) https://orcid.org/0000-0002-2552-6725
Koichi Kawakami (iD) http://orcid.org/0000-0001-9993-1435
Kei Yamamoto (iD) https://orcid.org/0000-0002-8775-9022

## Ethics

Animal experimentation: The experimental protocols and care of laboratory animals were conducted in compliance with the official regulatory standards and approval of the French Government (reference document n°APAFIS#1286- 2015062616102603 v5), the official Japanese regulations for research on animal, and the regulations on Animal Experiments in Nagoya University.

## Decision letter and Author response
Decision letter https://doi.org/10.7554/eLife.54945.sa1
Author response https://doi.org/10.7554/eLife.54945.sa2

# Additional files
## Supplementary files
• Supplementary file 1. Conditions of tamoxifen treatments. A summary table showing experimental conditions of tamoxifen treatments in two quadruple transgenic lines: *Tg(her5:ERT2CreERT2;βactin: lox-stop-lox-hmgb1-mCherry;279A-GFP)* and *Tg(Dr830:ERT2CreERT2;βactin:lox-stop-lox-hmgb1-mCherry;279A-GFP)*. For the latter transgenic line, the conditions were adapted depending on the developmental stages.

• Transparent reporting form

## Data availability

Source data files for a transgenic zebrafish line have been provided online: https://tel.archives-ouvertes.fr/tel-01968023. All the other data generated or analysed during this study are included in the manuscript and supporting files.

The following previously published dataset was used:

| Author(s) | Year | Dataset title | Dataset URL | Database and Identifier |
|---|---|---|---|---|
| Kawakami K, Abe G, Asada T, Asakawa K, Fukuda R, Ito A, Lal P, Mouri N, Muto A, Suster ML, Takakubo H, Urasaki A, Wada H, Yoshida M | 2010 | Zebrafish Gene Trap and Enhancer Trap Database | http://kawakami.lab.nig. ac.jp/ztrap/ | zTrap database: gSAGFF279A, gSAGFF279A |

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

# Appendix 1

**Appendix 1—key resources table**

| Reagent type (species) or resource | Designation | Source or reference | Identifiers | Additional information |
|---|---|---|---|---|
| Strain, strain background (*Danio rerio*) | *Tg(gSAGFF279A; UAS:GFP)* | This study | zTrap database: gSAGFF279A | transgenic zebrafish |
| Strain, strain background (*Danio rerio*) | *Tg(UAS:GFP)* | **Asakawa and Kawakami, 2009** | zTrap database: UASGFP1A | transgenic zebrafish |
| Strain, strain background (*Danio rerio*) | *Tg(her5: ERT2CreERT2)* | **Galant et al., 2016** | | transgenic zebrafish |
| Strain, strain background (*Danio rerio*) | *Tg(Dr830: ERT2CreERT2)* | **Heuzé, 2017** | | transgenic zebrafish |
| Strain, strain background (*Danio rerio*) | *Tg(βactin:lox-stop-lox-hmgb1-mCherry)* | **Wang et al., 2011** | | transgenic zebrafish |
| Antibody | chicken polyclonal anti-GFP | Aves Labs | Cat# GFP-1020; RRID:AB_10000240 | 1:1000 (brain sections), 1:400 (whole brain) |
| Antibody | rabbit polyclonal anti-dsRed | Takara Bio | Cat# 632496; RRID:AB_10013483 | 1:600 |
| Sequence-based reagent | oligonucleotide *ert2* forward primer | Eurogentec (sequence based on **Dirian et al., 2014**) | | 5'-ATGGCCGGTGACATGAGAGCTG-3' |
| Sequence-based reagent | oligonucleotide *Cre* reverse primer | Eurogentec (sequence based on **Dirian et al., 2014**) | | 5'-CATCAGGTTCTTCCTGACTTCAT-3' |
| Chemical compound, drug | tamoxifen | Sigma-Aldrich | T176 | |
| Software, algorithm | Amira 6.5.0 | Thermo Fisher Scientific-FEI | | |
| Software, algorithm | Imaris 8.0.1 | Oxford Instruments | | |
| Software, algorithm | Fiji / ImageJ | **Schindelin et al., 2012** | | http://fiji.sc/ |
| Software, algorithm | ImageJ cell counter plugin | Kurt De Vos (University of Sheffield) | | https://imagej.nih.gov/ij/plugins/cell-counter.html |
| Other | biocytin | Sigma-Aldrich | B4261 | |
| Other | biotinylated dextran amine (BDA) | Thermo Fisher Scientific | D7135 | |
| Other | DiI | Thermo Fisher Scientific | D7756 | |
| Other | DiD | Thermo Fisher Scientific | L7781 | |
| Other | DAPI | Sigma-Aldrich | 32670 | |

