## [Decision Letter]

**Acceptance summary:**

In this manuscript, the authors convincingly show that PGl and PGa contain numerous mCherry+ cells showing that they are derived from the mesencephalon, although it remains unlikely that the entire preglomerular complex (PG-C) is mesencephalon-derived. These findings are important for understanding the mechanism of forebrain evolution.

**Decision letter after peer review:**

Thank you for submitting your article "Non-thalamic origin of zebrafish sensory relay nucleus: convergent evolution of visual pathways in amniotes and teleosts" for consideration by *eLife*. Your article has been reviewed by three peer reviewers, and the evaluation has been overseen by Marianne Bronner as the Senior and Reviewing Editor. The following individual involved in review of your submission has agreed to reveal their identity: Glenn Northcutt (Reviewer 3).

The reviewers have discussed the reviews with one another and the Reviewing Editor has drafted this decision to help you prepare a revised submission.

Summary:

In this manuscript, the authors provide developmental evidence that many, if not all, of the pallial projecting cells of the preglomerular complex are of midbrain origin, and conclude that this is a case of convergent evolution between teleosts and other vertebrates. The reviewers found the manuscript interesting but also suggest numerous revisions to improve the manuscript. Rather than enumerating them, please see the detailed reviews below. We believe these will greatly improve the manuscript and can be accomplished in a reasonable period of time.

Reviewer 1:

The authors present valuable new data that address a longstanding question in comparative neuroanatomy, namely whether the visual pathway ascending from the optic tectum to the telencephalon in teleosts passes through the dorsal thalamus, as it does in amniotes. The authors present convincing evidence that (at least) a substantial number of the "relay" neurons in this pathway develop from mesencephalic precursors, rather than from the diencephalon/thalamus. This is an important finding in part because it strongly suggests that the "tectofugal" visual pathway in teleosts is not homologous to that in amniotes. I'm convinced by the data but see two potential weaknesses:

1) In each individual fish (which were tamoxifen-treated at different stages of development), only a relatively small fraction of the cells labeled for mesencephalic origin are double-labeled with the ascending PG neuron marker (i.e., the "relay" neuron marker). The authors argue that this double-labeling percentage is low because the tamoxifen induction (for the Cre-based lineage tracing) is limited to a relatively short period of time and the Cre-lox system can, in any case, never label 100% of the neurons. It's a bit embarrassing to admit, but I am uncertain whether this argument (paragraph one of the Discussion) is valid (shouldn't one nonetheless expect more double-labeled cells after tamoxifen induction in early development?). Fortunately, the editor will surely be in a better position to make this judgment than I am. The question is important because the work may well be criticized on the grounds that "not all" of the relay (i.e., PG) neurons were shown to originate from the mesencephalon, leaving open the possibility that some are thalamic. The authors themselves seem rather tentative as well ("it may be reasonable to assume that most GFP+ cells originate from the midbrain").

2) A key phyogenetic argument is muddled. Specifically the final paragraph of subsection “Evolution of ascending visual pathways” is very confusing or even misleading. I think it would be best simply to write something like: "Substantial tectofugal visual pathways to the pallium are observed only in amniotes and teleosts (forget the "Osteichtyes" reference; it will only confuse most readers) and not in the intermediate taxa. In addition, the data in the present paper clearly indicate that many (though perhaps not all) of the PGl neurons in the teleost pathway are (in contrast to amniotes) not part of the thalamus, but instead have a mesencephalic origin. In combination, these observations strongly suggest that the teleost and amniote pathways are not homologous to one another (i.e. have evolved independently of one another)." The business about the ancestor of bony fishes having had "diffuse visual pathways" is not well justified. I'm not even sure what it's supposed to mean. Perhaps the authors envision some sort of differential elaboration from an ancestral diffuse state, akin to Ebbesson's old idea of parcellation, but there is no evidence for that and, surely, de novo pathways are not a logical impossibility. Easily fixed, though.

Reviewer 2:

Combining tamoxifen inducible cre-lox p fate mapping (lineage tracing) with tract tracing methodology in a transgenic line of the teleost zebrafish, the authors claim to show that the alar plate mesencephalon (tectum opticum) contributes perhaps the majority neurons to a lateral teleostean visual relay nucleus of the preglomerular complex (PGl) located in a laterally displaced diencephalic territory. The authors performed several experiments and incubated over various time frames to show that many neurons if not the majority of the PGl derives from the tectum opticum. In addition, the study shows that specifically these GFP-positive mesencephalic neurons project into those two teleostean pallial territories, the main projection area is the dorsolateral pallium (Dl) and a minor projection area is located in the dorsomedial pallium (Dm). The authors discuss their findings regarding the evolution of homologs and non-homologs to the mammalian thalamo-cortical projections specifically the mammalian geniculate and extra-geniculate pathways. Based on the results, the authors conclude that zebrafish/teleosts preglomerular complex is independently evolved "thalamus-like" relaying center and the PGl specifically is a mesencephalic nucleus that cannot be compared to neither the geniculate nor the extrageniculate pathway. Thus, the study provides important new information to the evolutionary origin of the PGl suggesting that the PGl projections to Dl and Dm arise from these mesencephalic GFP positive neurons.

Overall, the manuscript is well written, the method of cre-loxp mediated fate mapping is advanced, and the results interesting. However, the overall enthusiasm about the findings is dampened because of issues related to how the findings are presented and that previous publications are either not properly or not at all cited. In addition, I have doubts that the PGl should be treated in its entirety as a mesencephalic entity. Instead, I would think that it's rather likely that the mesencephalon contributes many cells to the PGl, which itself resides in the diencephalon. It looks like what the authors describe should be considered tangential migration into the diencephalon. My recommendation for publication, thus, would be dependent upon revisions and reactions from the authors towards these criticisms.

Criticisms (Overview):

Most importantly, I would like to see more cre-loxP staining that show the development of the PGl and migrating GFP positive neurons between the TeO and its supposedly final destination the PG.

There are major problems with regard to the introduction, rationale, and discussion of the evolution of visual thalamo-cortical or thalamo-pallial like projections in teleosts and their comparability to mammals (and/or birds). Here, the authors avoid discussing how the zebrafish forebrain and in particular the everted pallium can be compared to (a) the overall prosomeric organization of the vertebrate brain (e.g., Puelles and Rubenstein, 2003, Puelles and Rubenstein, 2015) and (b) with regard to several competing pallial eversion theories that postulate different locations for the homologs of the dorsal (mammalian iscortex) and medial pallial (hippocampus) territories. With regard to the projections of the PGl: Former studies (cited by the authors) already established, that PGl in teleosts project to two pallial territories of the teleostean telencephalon, the medial (Dm) and dorsolateral (Dl) zones most often respectively considered the homologs of the pallial amygdala and medial (hippocampal) pallium (based on prox1 gene expression, Ganz et al., 2012). However, Dl (together with a central area (Dc) and dorsal territory (Dd) also have been discussed as being homologous or functionally equivalent to the mammalian dorsal pallium (isocortex). The authors need to state how the interpret homologies and based on what criteria (developmental as is the case for the PGl?). The putative homologies to the mammalian dorsal pallium, amygdala and hippocampuse need to be discussed in order to make sense out of the comparison to the mammalian thalamo-cortical projections, because if there was no distinct isocortex-homolog in teleosts at the first place, the postulation of potentially homologous thalamo-cortical projections would be pointless. Basically, the authors need to declare their position on where the dorsal pallium homolog (if they think it exists) is located in the everted pallium of zebrafish. To my understanding, the authors define homology first and foremost by determination of the topological place of origin, whereas function or connectivity can change during evolution as similarly pointed out by Nieuwenhuys (2009). It is unclear to the reader if they also define pallial homologies based on developmental criteria. The schematics (specifically Figure 1) that describe the visual pathways are way too simplistic.

My recommendation is that these issues need to be solved before considering publishing in *eLife*.

In the following, I will detail the problems.

1) Lack of citations and balanced discussion/.

First of all, the idea that the preglomerular complex is an independently (parallel) evolved thalamus-like structure is not new and the authors should give credit to some publications that have expressed this hypothesis before. Most obvious, there is a relatively recent review about the zebrafish thalamus organization (Mueller, 2012) that already states that the PG-complex should be treated as an independently evolved thalamus-like structure (similar to some studies before cited in that review). The review also focuses on the comparison of the PGl connectivity with regard to the mammalian geniculate and extra-geniculate pathways with a prominent figure that outlines that the zebrafish pathways are not homologous to these mammalian pathways. Although most previous works considered the PG-complex mainly a derivative of the posterior tuberculum (e.g. the early works of the Northcutt group some of which are cited by the authors), developmental studies on proliferation patterns using PCNA and BrdU already indicated that the PG-complex is of "multiprosomeric origin" and that even mesencephalic neurablasts migrate into the developing PG-complex (Wullimann and Puelles, 1999, Mueller and Wullimann, 2002). It is hard to imagine that the authors have not recognized these studies as they are also described in depth in the atlas of the developing zebrafish brain (Mueller and Wullimann, 2005, 2nd edition 2016). Wullimann and Mueller also provided alternative eversion hypotheses (Mueller and Wullimann, 2009) on the location and definition of the dorsal pallium (Mueller et al., 2011, Mueller, 2012). Even though the authors most likely disagree with these alternative views, they cannot simply ignore them. They need to discuss them reasonably balanced in comparison with their own ideas (e.g., Ito and Yamamoto, 2009, Yamamoto et al., 2007) to illustrate their view on pallial homologies. For example, in their complex eversion model of 2007, Yamamoto et al. postulated that many different areas of the pallium (dorsomedial (Dm), dorsolateral (Dl), dorsocentral (Dc), and dorsal domain (Dd) constitute the teleostean homolog of the dorsal pallium. To be clear, I am not suggesting that the authors need to discuss eversion models in detail, but they at least need to mention that there are diverging interpretations on pallial organization and that this is an open question or that comparative anatomists have not reached a consensus on that question.

2) Prosomeric organization not considered and incorrect citations.

The next problem concerns the potentially multi-prosomeric organization of the zebrafish PG-complex (PGC). Here the authors too often switch between the PGl, which they found to be a derivative of the mesencephalon, and the PG-complex at large. Here the authors fail again to provide a nuanced perspective on the matter at hand. So, basically the authors do not address the problem that the remaining nuclei of the PGC are obviously not derived from the tectum opticum nor do these remaining PG nuclei seem to show migrated GFP positive TeO cells, which in turn leaves open the possibility that each single nucleus is derived from only one or from multiple different prosomeric or other domains. Again, the authors need to develop a more balanced view on the organization and potential homologies to mammalian or tetrapod forebrains and cite more previously published articles in a balanced manner. In addition, it would be nice if they compared the PGl with any other nucleus of the PG-complex with a potentially different place of origin (if they think that single nuclei have very distinct developmental origins? But this is of course just a suggestion to broaden the hypothesis which otherwise would need to be restricted to the PGl).

For example:

The authors write that "some studies suggest that Pax6-positive (Pax6+) alar cell population migrate ventrally to form the PG during development (Ishikawa et al., 2007;Wullimann and Rink, 2001), claiming that the teleost PG is homologous to the tetrapod thalamus".

This statement is incorrect at least for Wullimann and Rink (2001, and 2002). Specifically, these authors clearly state already in their Abstract (2002) that: "The peripherally migrated, adult diencephalic preglomerular complex of the basal plate posterior tubercle (early: M2) provides sensory inputs to the pallium. Early Pax6 protein distribution indicates that at least part of M2 may directly originate from alar plate ventral thalamic Pax6-expressing cells." In other words, their work suggests that ventral thalamic (or prethalamic) pax6 cells contribute to the PG complex and not that the PG complex as a whole is homologous to the thalamus proper (dorsal thalamus).

Again, the idea that the PG complex is of multiprosomeric origin (Wullimann and Puelles, 1999) already implies that it cannot be fully homologous, in theory only portions of it or single nuclei can be homologous or field homologous. The authors should not interchangeably compare thalamus with prethalamus, there is a huge difference between these entities.

Reviewer 3:

Although the preglomerular nucleus of teleosts is known to receive multiple ascending sensory projections, and to project to the pallium, its homology to cell groups in other vertebrates is uncertain. It has been claimed to be homologous to parts of the dorsal thalamus (Yamoto and Ito, 2008) and to the posterior tubercle (Striedter and Northcutt, 2020) In this manuscript, the authors provide developmental evidence that many, if not all, of the pallial projecting cells of the preglomerular complex are of midbrain origin , and they conclude that this is a case of convergent evolution between teleosts and other vertebrates. The evidence appears convincing and would represent resolution of an important topic in vertebrate brain evolution.

[Editors' note: further revisions were suggested prior to acceptance, as described below.]

Thank you for submitting your article "Non-thalamic origin of zebrafish sensory nuclei implies convergent evolution of visual pathways in amniotes and teleosts" for consideration by *eLife*. Your article has been reviewed by two peer reviewers, and the evaluation has been overseen by Marianne Bronner as the Senior Editor. The reviewers have opted to remain anonymous.

The reviewers have discussed the reviews with one another and the Reviewing Editor has drafted this decision to help you prepare a revised submission.

Summary:

In this manuscript, the authors convincingly show that PGl and PGa contain numerous mCherry+ cells showing that they are derived from the mesencephalon, although it remains unlikely that the entire preglomerular complex (PG-C) is mesencephalon-derived. These findings are important for understanding the mechanism of forebrain evolution.

Revisions:

As detailed in the reviews below, the reviewers raise many points that can be addressed by careful changes to the text. For example, while the overall readability of the manuscript has improved, t it would be important to add more in-depth discussion about alternative views and data that speak against their hypothetisis. Although these and other changes are editorial, they are also extremely critical for correct presentation of the data so are more than minor. I refer you to the complete reviews below for further details.

Reviewer 1:

The authors have nicely addressed my concerns and, as far as I can tell, the main concerns of the other reviewers. I do like the new/additional figures and am convinced by them. I only have four substantial (but not major) recommendations (plus some more minor editorial suggestions). Overall, I think it's a strong paper that makes an important contribution to its field.

Introduction paragraph three, opening sentence: This sentence is flawed. Instead, I suggest: "The principal problem is that it is difficult to determine the primitive condition for the thalamopallial pathways of amniotes."

Discussion paragraph four: I suggest, instead: "The earlier data had been interpreted as demonstrating a forebrain origin of PG cells, implying that he Pax6+ cells are thalamic or prethalamic. Our data, in contrast, show that a significant percentage of PG cells are of mesencephalic origin." I would leave the tectal cells out of it, despite what one of the other reviewers wrote; I think the figures show clearly enough that, yes, many tectal cells are labeled as well.

Paragraph three subsection “Evolution of ascending visual pathways”: I suggest: "Our study does not solve the pallial homology problem, but it strongly suggests that the tectofugal pathways in amniotes and teleosts are not homologous to one another. In tetrapods, the neurons giving rise to the pallial projections all have their cell bodies in the thalamus, even though their pallial projection targets differ between the major amniote groups."

Figure 8—figure supplement 2: for the y-axis label, I suggest: "mCherry+ cells (cumulative %)”; and for the legend, I suggest: "Cherry+ cells observed after tamoxifen induction at the indicated time points" and "mCherry+ cells observed after induction at earlier developmental stages"

Reviewer 2:

After reading the revised manuscript and response letter of the authors, I recommend some minor revisions before considering the article for *eLife*. The authors now convincingly show that PGl and PGa contain numerous mCherry+ cells, however I still find it unlikely that the entire preglomerular complex (PG-C) is a derivative of the mesencephalon. Since the overall readability after revision has considerably increased, I do better understand the authors intents though and both the introduction and results part have been improved a lot. Within the discussion part, they could discuss somewhat more in depths alternative views and data that speak against their own hypothetical scenario. Their findings are important for understanding the mechanism of forebrain evolution. To be precise, my previously expressed major concern is that developmental gene expression patterns, BrdU-experiments, shh-driven GFP, and GFAP-radial glia fiber distribution speak in favor of a radial migration from alar and basal plates of the prosomeric forebrain plus some either radial or tangential contributions of the mesencephalon) for most of the preglomerular complex. It is incorrect in this context to state that former studies on the developmental origin of the PG-C focused "only" on molecular distributions, as the authors want to imply. There is a whole body of work that suggest a multiprosomeric orgin including some (tangential) migration originating in the alar plate mesencephalon.

The controversy or problem could simply be solved if the authors would discuss their evolutionary hypothesis in a slightly more considerate way and in comparison to some of the papers that suggested a prosomeric organization (see further below). This would make the paper stronger in the end, not weaker. They do find that a huge number of mesencephalic neurons contribute to the preglomerular complex. As their results, however, do not allow to determine if the mCherry+ cells are the results of tangential or radial migration, they need also to make a statement here about what they think is happening.

In the following, I will respond to their rebuttal letter in some detail:

Point 1) Validation whether the entire PG is of the mesencephalic origin (mCherry+)

Authors:

As we have already mentioned, the percentage of mCherry+ in each induction is not very high, around 10-20%. However, it has to be considered that the PG continues to develop up to 6-8 wpf, thus the induction time represents only a fraction of the developmental timeframe. Based on these data, it would be reasonable to conclude that the majority (if not all) of PG is constituted by mesencephalic neurons.

Reviewer 2:

The authors need to be more careful with their assumptions, if they can induce only about 10 to 20% and also did not study the entire timeframe of PG development. Plus, there is probably also some apoptosis ongoing that these authors do not take into consideration. Although the PG continuous to develop into 6 to 8 wpf, the proliferation and migration activity has probably slowed down considerably compared to the early timeframe. Plus, the authors themselves do in fact state something similar in their Discussion.

The authors state:

“It is difficult to prove whether all the PG cells originate from the mesencephalon, due to the technical limitation of the tamoxifen induction. Long term tamoxifen treatment leads to a high mortality rate of the fish during the experiment (Bloch et al., 2019; Yu et al., 2019). Moreover, the Cre-lox system would not allow 100% induction rate (Hayashi and McMahon, 2002). Thus, mCherry labeling of each experiment would represent only a fraction of the cells originating from the mesencephalic region.”

So, how can the authors state that it is difficult to proof a claim and then act as if they did proof it?

In addition, and as I stated before, numerous studies on GFAP and other molecular distributions including shh-driven GFP, distribution of GABAergic neurons indicated a prosomeric organization of the zebrafish forebrain and the PG-C is most likely of multiprosomeric origin including contributions from the mesencephalon as stated before (Puelles and Wullimann, 1999, Mueller and Wullimann, 2004, etc.). Again, the authors did not provide evidence or thoroughly discussed the problem of tangential versus radial migration. The problem of radial versus tangential migration is pivotal for both discussing potential homologies and correctly interpreting the topological origin of the preglomerular complex. This should be discussed in one paragraph in the Discussion.

Point 2) Discussion on the pallial homology in the context of eversion theory.

Authors:

As pointed out, we here discuss the "regional homology", based on developmental criteria. Recent publications based on cell lineage studies (Dirian et al., 2014; Furlan et al., 2017) have indicated that the classical eversion theory needs to be modified. We have discussed this point in our previous publications, proposing a new view on the pallial homology (Yamamoto et al., 2017; Yamamoto and Bloch, 2017).

Reviewer 2:

I am okay with not discussing pallial homology, but I suggest citing also the Folgueira et al., 2012 paper about the morphogenesis underlying the development of the everted teleost telencephalon, because it is significance for understanding the eversion process.

3) Discussion and references on previous works in relation to the prosomeric model

Author's statement in response to previous comments of reviewer 2.

We do not agree to use expression of genes as the only argument to define brain regions, thus we see problems on the conclusions of some previous works proposed to refer by reviewer 2. Our position has been discussed in our previous publications (Affaticati et al., 2015; Yamamoto et al., 2017; Yamamoto and Bloch, 2017), and we mentioned it briefly in Introduction.

However, we avoid to discuss in detail because it would not be directly relevant to the topic here. Again, regardless of the position concerning the prosomeric model, our findings favors the hypothesis that the PG, as a whole, is not homologous to the amniote thalamus.

Reviewer 2:

The authors oversimplify by stating that reviewer 2 argues to "use expression of genes as the only argument to define brain regions." The prosomeric model is currently the established paradigm in comparing forebrain organization across vertebrates and a lot of former studies on developmental gene expressions, developmental and adult GFP progeny studies in transgenic lines do in fact support an overall prosomeric organization of the forebrain in teleosts as in other vertebrates. I understand that the authors aim to promote a new "model" or "theory" of forebrain evolution. However, as their model is not widely accepted in the field and the broader audience of *eLife*, of course, they need to discuss the established or current standard model(s) somewhat more in depth. Last but not least, it would probably help to make their hypothesis stronger.

I am okay with some of the other statements and the authors' , so I will not go into further detail with regard to their responses here.

Regarding more cre-loxP….

The authors write:

Please note that the GFP does not represent the PGl cell identity. The expression of GFP can be switched on and off, depending the regulation of the inserted locus (inpp5ka gene). We think this is the case here. The GFP may be expressed only after maturation (the PGl cells arrive at the final destination). Thus the experiment prosed by reviewer 2 would not provide the information that he or she expects.

Reviewer 2:

Okay, it seems as if fate mapping has some weaknesses in zebrafish, and I see this as an argument to be more cautious with regard to the estimated numbers of cells reaching PGl, and PG at large.

Evaluation of homology.

I accept the changes and do not have further comments here.

Lack of citations, the authors state:

We added this reference to the Introduction. It is true that hypotheses by Mueller, 2012, on the ascending auditory and visual pathways (Figure 7 and 8) suggest that they are not homologous to the thalamocortical pathways. However, we find a problem on the rationale: the author claims that the PG-Dl projection is not homologous to the corticothalamic projection because the entire Dl should be homologous to the hippocampus. Thus, we cited the position of the author rather in the context of the controversy of pallial homology.

Reviewer 2:

This is not entirely true, because Mueller also considered developmental data and gene expression analyses in line with the Wullimann papers (e.g. Wullimann and Mueller, 2002) and the distribution of gad67 mRNA in the adult brain (Mueller and Guo, 2009). This latter paper would also need to be cited in the context of what Wullimann describes with the shh-GFP line, that is, that distributions of molecular markers are radially organized suggesting a prosomeric ground plan of the diencephalon that includes the PG-C in their view.

I am not saying that the authors have to agree with any of these alternative views, but it would make the discussion stronger and it would be helpful for future discussions.

The authors write:

We admit that we often switch PGl and PG. In the sentence discussing about the "visual" projection neurons, we refer PGl, whereas when the point of discussion concerns the entire PG or "sensory" projections in general, we tend to refer PG.

In the previous version, we did not clearly describe that the mCherry+ cells are distributed throughout the PG (other than PGl), which may have led the confusion. We here provide new figures (new Figure 5 and Figure 5—figure supplement 1) to demonstrate that the mCherry+ cells are not limited to the PGl, but also found other nucleus of the PG (such as PGa).

To avoid the confusion, we modified the "PGl" to "PG" in some figures and descriptions, unless it is important to exclude other nuclei of PG.

Reviewer 2 response:

The newly added Figure 5 definitely helps to convince the reader that there are substantial amounts of mCherry+ cells in both PGl and PGa. The readability has improved a lot. However, to truly convince the readers that the entire PG-C is a derivative of the mesencephalon photographs are needed that also show mCherry+ cells are equally distributed in all of nuclei of the PG-C. So, the authors should also show a microphotograph of the PGm, for example, and all other nuclei of the PG-C that they consider be derivatives of the mesencephalon. Based on the photographs they already provided, it should be easy to show also close-up photographs of mCherry+ neurons in the remaining PG nuclei.

With regard to the remaining points I raised in my last review, I am satisfied with the changes made by the authors. In addition, I am also globally okay with the text changes. In the following, I will only make some comments specific to the altered manuscript.

The authors write:

“The her5-mCherry+ cells are distributed in the entire PG, including the PGl where the 279A-GFP+ pallial projection neurons are located (Figure 5 and Figure 5—figure supplement 1). Thus, we conclude that many of PG cells, if not all, are of the midbrain

origin.”

Reviewer 2 comments:

Figure 5 shows mCherry+ cells nicely in the inlet of 5E and PGl and PGa. Could the author also label the other PG nuclei showing such a cross sections? For example, PGm should be shown as well as all other nuclei that they consider derivatives of the mesencephalon. Globally stating that the entire PG is a derivative is not precise enough.

Discussion paragraph four:

The authors do now discuss in a well-balanced manner the hypotheses regarding pax6-positive and shh-positive cell contributions to the PG-C in lines.

However, a paragraph is missing that discusses the findings that PCNA, and longterm BrdU-studies in comparison to expression patterns of ngn1 and neuroD indicated contributions from both alar and basal plate diencephalic territories (Pretectum, Thalamus, PTd, PTv) to the PG-C (M2) between 2 and 5 dpf in larval zebrafish (Wullimann and Puelles, 1999, and Mueller and Wullimann, 2002). This is important and cannot just simply ignored in the discussion because the PG-C is probably not only comprised of ex-pax6-positive neurons coming from the mesencephalon. It is unlikely, that these ngn1/neuroD positive neurons are shh- or pax6 positive. So, besides pax6 and shh-positive contributions, the authors cannot rule out that glutamatergic precursors from these diencephalic territories contribute to the PG-C.

---

## [Author Response]

Reviewer 1:The authors present valuable new data that address a longstanding question in comparative neuroanatomy, namely whether the visual pathway ascending from the optic tectum to the telencephalon in teleosts passes through the dorsal thalamus, as it does in amniotes. The authors present convincing evidence that (at least) a substantial number of the "relay" neurons in this pathway develop from mesencephalic precursors, rather than from the diencephalon/thalamus. This is an important finding in part because it strongly suggests that the "tectofugal" visual pathway in teleosts is not homologous to that in amniotes. I'm convinced by the data but see two potential weaknesses:1) In each individual fish (which were tamoxifen-treated at different stages of development), only a relatively small fraction of the cells labeled for mesencephalic origin are double-labeled with the ascending PG neuron marker (i.e., the "relay" neuron marker). The authors argue that this double-labeling percentage is low because the tamoxifen induction (for the Cre-based lineage tracing) is limited to a relatively short period of time and the Cre-lox system can, in any case, never label 100% of the neurons. It's a bit embarrassing to admit, but I am uncertain whether this argument (paragraph one of the Discussion) is valid (shouldn't one nonetheless expect more double-labeled cells after tamoxifen induction in early development?). Fortunately, the editor will surely be in a better position to make this judgment than I am. The question is important because the work may well be criticized on the grounds that "not all" of the relay (i.e., PG) neurons were shown to originate from the mesencephalon, leaving open the possibility that some are thalamic. The authors themselves seem rather tentative as well ("it may be reasonable to assume that most GFP+ cells originate from the midbrain").

In a previous publication, we have already demonstrated that many PG cells are derived from the mesencephalon (Bloch et al., 2019). The PG develops as the anterior end of the migrating MHB progenies, and the entire region containing the PG is labeled with mCherry. In the revised manuscript, we provide additional data to demonstrate it (new Figure 5, Figure 5—figure supplement 1, Figure 8—figure supplement 2, and Video 2). Thus, even without the double-labeling data (co-localization of GFP and mCherry), we can reasonably assume that the majority of PG cells are of mesencephalic origin. The GFP/mCherry double labeling further confirm that the pallial projection neurons are not exceptional, and they also are derived from the mesencephalon. Altogether, it is reasonable to conclude that the majority of PG cells, including the "thalamic-like" projection neurons, are of mesencephalic, not thalamic origin.

In Figure 8—figure supplement 2, we provide an estimation of the proportion of mCherry+ cells. This indicates that around 60% of PG cells observed in the adult originate from the midbrain between 24hpf and 3 wpf. We have demonstrated that the mCherry+ in PG would continue to increase up to around 6-8 wpf. Thus it would be reasonable to conclude that the majority (if not all) of PG cells are constituted by mesencephalic neurons.

2) A key phyogenetic argument is muddled. Specifically the final paragraph of subsection “Evolution of ascending visual pathways” is very confusing or even misleading. I think it would be best simply to write something like: "Substantial tectofugal visual pathways to the pallium are observed only in amniotes and teleosts (forget the "Osteichtyes" reference; it will only confuse most readers) and not in the intermediate taxa. In addition, the data in the present paper clearly indicate that many (though perhaps not all) of the PGl neurons in the teleost pathway are (in contrast to amniotes) not part of the thalamus, but instead have a mesencephalic origin. In combination, these observations strongly suggest that the teleost and amniote pathways are not homologous to one another (i.e. have evolved independently of one another)." The business about the ancestor of bony fishes having had "diffuse visual pathways" is not well justified. I'm not even sure what it's supposed to mean. Perhaps the authors envision some sort of differential elaboration from an ancestral diffuse state, akin to Ebbesson's old idea of parcellation, but there is no evidence for that and, surely, de novo pathways are not a logical impossibility. Easily fixed, though.

We have modified the description.

Reviewer 2:Combining tamoxifen inducible cre-lox p fate mapping (lineage tracing) with tract tracing methodology in a transgenic line of the teleost zebrafish, the authors claim to show that the alar plate mesencephalon (tectum opticum) contributes perhaps the majority neurons to a lateral teleostean visual relay nucleus of the preglomerular complex (PGl) located in a laterally displaced diencephalic territory. The authors performed several experiments and incubated over various time frames to show that many neurons if not the majority of the PGl derives from the tectum opticum. In addition, the study shows that specifically these GFP-positive mesencephalic neurons project into those two teleostean pallial territories, the main projection area is the dorsolateral pallium (Dl) and a minor projection area is located in the dorsomedial pallium (Dm). The authors discuss their findings regarding the evolution of homologs and non-homologs to the mammalian thalamo-cortical projections specifically the mammalian geniculate and extra-geniculate pathways. Based on the results, the authors conclude that zebrafish/teleosts preglomerular complex is independently evolved "thalamus-like" relaying center and the PGl specifically is a mesencephalic nucleus that cannot be compared to neither the geniculate nor the extrageniculate pathway. Thus, the study provides important new information to the evolutionary origin of the PGl suggesting that the PGl projections to Dl and Dm arise from these mesencephalic GFP positive neurons.Overall, the manuscript is well written, the method of cre-loxp mediated fate mapping is advanced, and the results interesting. However, the overall enthusiasm about the findings is dampened because of issues related to how the findings are presented and that previous publications are either not properly or not at all cited. In addition, I have doubts that the PGl should be treated in its entirety as a mesencephalic entity. Instead, I would think that it's rather likely that the mesencephalon contributes many cells to the PGl, which itself resides in the diencephalon. It looks like what the authors describe should be considered tangential migration into the diencephalon. My recommendation for publication, thus, would be dependent upon revisions and reactions from the authors towards these criticisms.

There are three major issues raised by reviewer 2:

1) Validation whether the entire PG is of the mesencephalic origin (mCherry+).

We provided additional data (new Figure 5, Figure 5—figure supplement 1, Figure 8—figure supplement 2, and Video 2) with modification of the corresponding text.

a) In order to demonstrate clearly that the entire PG develops from the mesencephalon, we added figures to show the progress of mCherry+ cells after induction at 24 hpf in

*Tg(her5:ERT2CreERT2;βact:lox-stop-lox-hmgb1-mCherry)*.

They are shown in the new Figure 5, Figure 5—figure supplement 1, and Video 2.

These images show that the mCherry+ cells are not limited to the nucleus corresponding to PGl, but distributed throughout in PG.

b) We also quantified the proportion of mCherry+ cells (Figure 8—figure supplement 2). The number of mCherry+ cells was counted in the brains induced at different time points (at 24 hpf, 48 hpf, 2 wpf, and 3 wpf). We provide an estimate that around 60% of PG cells observed in the adult originate from the midbrain between 24hpf and 3 wpf.

As we have already mentioned, the percentage of mCherry+ in each induction is not very high, around 10-20%. However, it has to be considered that the PG continues to develop up to 6-8 wpf, thus the induction time represents only a fraction of the developmental timeframe. Based on these data, it would be reasonable to conclude that the majority (if not all) of PG is constituted by mesencephalic neurons.

2) Discussion on the pallial homology in the context of eversion theory.

We have re-organized the Introduction so that the reader can better follow our point of view. We also mentioned that there are debates on the pallial homology.

As pointed out, we here discuss the "regional homology", based on developmental criteria. Recent publications based on cell lineage studies (Dirian et al., 2014; Furlan et al., 2017) have indicated that the classical eversion theory needs to be modified. We have discussed this point in our previous publications, proposing a new view on the pallial homology (Yamamoto et al., 2017; Yamamoto and Bloch, 2017).

We avoided to discuss the hypotheses of pallial homology in details, because it is not directly related to our conclusion and rather confusing for readers who are not familiar to all the controversy on the topic.

An important point of this article is that regardless of the pallial homology, the current data showing the non-homology between teleost PG and amniote thalamus strongly suggest that the PG-pallial projections in teleosts are not homologous to the thalamocortical projections in mammals. We re-organized a part of Discussion also.

3) Discussion and references on previous works in relation to the prosomeric model.

We do not agree to use expression of genes as the only argument to define brain regions, thus we see problems on the conclusions of some previous works proposed to refer by reviewer 2. Our position has been discussed in our previous publications (Affaticati et al., 2015;; Yamamoto et al., 2017; Yamamoto and Bloch, 2017), and we mentioned it briefly in Introduction.

However, we avoid to discuss in detail because it would not be directly relevant to the topic here. Again, regardless of the position concerning the prosomeric model, our findings favors the hypothesis that the PG, as a whole, is not homologous to the amniote thalamus.

Criticisms (Overview):Most importantly, I would like to see more cre-loxP staining that show the development of the PGl and migrating GFP positive neurons between the TeO and its supposedly final destination the PG.

Please note that the GFP does not represent the PGl cell identity. The expression of GFP can be switched on and off, depending the regulation of the inserted locus (inpp5ka gene). We think this is the case here. The GFP may be expressed only after maturation (the PGl cells arrive at the final destination). Thus the experiment prosed by reviewer 2 would not provide the information that he or she expects.

A short-term tracing experiment (tamoxifen induction at 4 wpf and observation several days after) has already been demonstrated in the previous publication (Bloch et al., 2019). Although not shown in the publication, we have also observed at 7 wpf (3 weeks after the induction at 4 wpf), which was similar to the observation at 3 mpf. Thus the progenies induced at 4 wpf seem to reach PGl within 3 weeks.

Instead of performing the experiment reviewer 2 suggested, we provided the new images to show a progress of mCherry+ cells along development in *Tg(her5:ERT2CreERT2;βact:lox-stop-lox-hmgb1mCherry)* after induction at 24 hpf.

[…] To my understanding, the authors define homology first and foremost by determination of the topological place of origin, whereas function or connectivity can change during evolution as similarly pointed out by Nieuwenhuys (2009). It is unclear to the reader if they also define pallial homologies based on developmental criteria. The schematics (specifically Figure 1) that describe the visual pathways are way too simplistic.

Here we have two criteria to evaluate non-homology of the tectofugal pathways: one is based on developmental data and the other is on cladistic analysis.

Developmental analysis: Regional homology can be tested by whether or not the two structures in different species are derived from the same topological position within the neural tube. Based on the developmental data, our results demonstrating the mesencephalic origin of many PG cells including the pallial projection neurons favors the hypothesis that PG is not homologous to the amniote thalamus.

Cladistic analysis: It is true that the connectivity pattern can change during evolution, but in that case, we expect to be able to reconstruct the evolutionary changes by cladistic analyses. If a characteristic between two species is homologous, we expect that many of the sister groups possess the characteristics.

This is not the case in the tectofugal pathways in bony vertebrates.

Thus, homology of tectofugal pathways between teleosts and mammals would be rejected at both levels of argument.

We revised the Discussion to better clarify this point.

My recommendation is that these issues need to be solved before considering publishing in eLife.In the following, I will detail the problems.1) Lack of citations and balanced discussion.First of all, the idea that the preglomerular complex is an independently (parallel) evolved thalamus-like structure is not new and the authors should give credit to some publications that have expressed this hypothesis before. Most obvious, there is a relatively recent review about the zebrafish thalamus organization (Mueller, 2012) that already states that the PG-complex should be treated as an independently evolved thalamus-like structure (similar to some studies before cited in that review).

We added this reference to the Introduction. It is true that hypotheses by Mueller, 2012, on the ascending auditory and visual pathways (Figure 7 and 8) suggest that they are not homologous to the thalamocortical pathways. However, we find a problem on the rationale: the author claims that the PG-Dl projection is not homologous to the corticothalamic projection because the entire Dl should be homologous to the hippocampus. Thus, we cited the position of the author rather in the context of the controversy of pallial homology.

Although most previous works considered the PG-complex mainly a derivative of the posterior tuberculum (e.g. the early works of the Northcutt group some of which are cited by the authors), developmental studies on proliferation patterns using PCNA and BrdU already indicated that the PG-complex is of "multiprosomeric origin" and that even mesencephalic neurablasts migrate into the developing PG-complex (Wullimann and Puelles, 1999, Mueller and Wullimann, 2002).

These references were cited in the Introduction.

We also added an image as the new Figure 5C, to show the primordium of PG that is comparable to the cell population reported in these publications (M" shown in Figure 7 of Wullimann and Puelles, 1999).

To be clear, I am not suggesting that the authors need to discuss eversion models in detail, but they at least need to mention that there are diverging interpretations on pallial organization and that this is an open question or that comparative anatomists have not reached a consensus on that question.

As suggested, we rewrote the Introduction to mention that there are controversies.

2) Prosomeric organization not considered and incorrect citations.The next problem concerns the potentially multi-prosomeric organization of the zebrafish PG-complex (PGC). Here the authors too often switch between the PGl, which they found to be a derivative of the mesencephalon, and the PG-complex at large. Here the authors fail again to provide a nuanced perspective on the matter at hand. So, basically the authors do not address the problem that the remaining nuclei of the PGC are obviously not derived from the tectum opticum nor do these remaining PG nuclei seem to show migrated GFP positive TeO cells, which in turn leaves open the possibility that each single nucleus is derived from only one or from multiple different prosomeric or other domains. Again, the authors need to develop a more balanced view on the organization and potential homologies to mammalian or tetrapod forebrains and cite more previously published articles in a balanced manner. In addition, it would be nice if they compared the PGl with any other nucleus of the PG-complex with a potentially different place of origin (if they think that single nuclei have very distinct developmental origins? But this is of course just a suggestion to broaden the hypothesis which otherwise would need to be restricted to the PGl).

We admit that we often switch PGl and PG. In the sentence discussing about the "visual" projection neurons, we refer PGl, whereas when the point of discussion concerns the entire PG or "sensory" projections in general, we tend to refer PG.

In the previous version, we did not clearly describe that the mCherry+ cells are distributed throughout the PG (other than PGl), which may have led the confusion. We here provide new figures (new Figure 5 and Figure 5—figure supplement 1) to demonstrate that the mCherry+ cells are not limited to the PGl, but also found other nucleus of the PG (such as PGa).

To avoid the confusion, we modified the "PGl" to "PG" in some figures and descriptions, unless it is important to exclude other nuclei of PG.

For example:The authors write that "some studies suggest that Pax6-positive (Pax6+) alar cell population migrate ventrally to form the PG during development (Ishikawa et al., 2007; Wullimann and Rink, 2001), claiming that the teleost PG is homologous to the tetrapod thalamus".This statement is incorrect at least for Wullimann and Rink (2001, and 2002). Specifically, these authors clearly state already in their Abstract (2002) that: "The peripherally migrated, adult diencephalic preglomerular complex of the basal plate posterior tubercle (early: M2) provides sensory inputs to the pallium. Early Pax6 protein distribution indicates that at least part of M2 may directly originate from alar plate ventral thalamic Pax6-expressing cells. " In other words, their work suggests that ventral thalamic (or prethalamic) pax6 cells contribute to the PG complex and not that the PG complex as a whole is homologous to the thalamus proper (dorsal thalamus).

We modified the statement in the Introduction, simply mentioning that "…(Pax6+) alar diencephalic cells migrate ventrally to form the PG".

In addition, we added new Figure 5C, in which the arrow indicates mCherry+ cells at the same level reported as the M" or M2 migrated preglomerular region shown in Wullimann and Puelles (1999). This reveals that at least there are migrated cells from the MHB in the M2 region.

[Editors' note: further revisions were suggested prior to acceptance, as described below.]

Reviewer 1:The authors have nicely addressed my concerns and, as far as I can tell, the main concerns of the other reviewers. I do like the new/additional figures and am convinced by them. I only have four substantial (but not major) recommendations (plus some more minor editorial suggestions). Overall, I think it's a strong paper that makes an important contribution to its field.

We thank reviewer 1 for his/her kind comments. Almost all the points raised by him/her are integrated in the text.

Reviewer 2:The problem of radial versus tangential migration is pivotal for both discussing potential homologies and correctly interpreting the topological origin of the preglomerular complex. This should be discussed in one paragraph in the Discussion.To be precise, my previously expressed major concern is that developmental gene expression patterns, BrdU-experiments, shh-driven GFP, and GFAP-radial glia fiber distribution speak in favor of a radial migration from alar and basal plates of the prosomeric forebrain plus some either radial or tangential contributions of the mesencephalon) for most of the preglomerular complex.

The main point raised by reviewer 2 and the editor is to discuss about alternative data that speak against our hypothesis. We thus added a paragraph about this point in Discussion.

Also, we mentioned that the hypothesis proposed by Wullimann and Rink (2001, 2002) are based on the prosomeric model (Wullimann and Puelles, 1999).

<bold />I am okay with not discussing pallial homology, but I suggest citing also the Folgueira et al., 2012, paper about the morphogenesis underlying the development of the everted teleost telencephalon, because it is significance for understanding the eversion process.

We added the reference in Introduction.

[….] (Mueller and Guo, 2009). This latter paper would also need to be cited in the context of what Wullimann describes with the shh-GFP line, that is, that distributions of molecular markers are radially organized suggesting a prosomeric ground plan of the diencephalon that includes the PG-C in their view.

We added the reference in Discussion.

To truly convince the readers that the entire PG-C is a derivative of the mesencephalon photographs are needed that also show mCherry+ cells are equally distributed in all of nuclei of the PG-C. So, the authors should also show a microphotograph of the PGm, for example, and all other nuclei of the PG-C that they consider be derivatives of the mesencephalon.

We added more posterior sections including PGm as Figure 5—figure supplement 1D and E.